# Leveraging Pretrained Knowledge at Inference Time: LoRA-Gated Contrastive Decoding for Multilingual Factual Language Generation in Adapted LLMs

**Gwangseon Jang**[1,3]**, Hongseok Choi**[2]**, Chanuk Lim**[3]**, Kyong-Ha Lee**[3,4]**, Mun Yong Yi**[1]*

[1]Graduate School of Data Science, KAIST, Daejeon, Republic of Korea
[2]Language Intelligence Research Section, ETRI, Daejeon, Republic of Korea
[3]Large-scale AI Research Center, KISTI, Daejeon, Republic of Korea
[4]University of Science and Technology (UST), Daejeon, Republic of Korea
{rayjang27, munyi}@kaist.ac.kr, hongking9@etri.re.kr
{gsjang, chanuklim, kyongha}@kisti.re.kr

## Abstract

Large language models (LLMs) adapted to specific languages through continual pretraining or instruction tuning often suffer from catastrophic forgetting, which can lead to factual inaccuracies. This issue is particularly pronounced in multilingual settings, where adaptation may override general world knowledge with language-specific patterns. We propose LoRA-Gated Contrastive Decoding (LGCD), a training-free inference-time decoding framework that improves factuality in language-adapted LLMs by leveraging knowledge from the original pretrained model. LGCD operates by (1) extracting factual representations from Feed-Forward Network (FFN) layers via LoRA-based decomposition, approximating pretrained knowledge, (2) dynamically gating decoding based on token-level confidence, and (3) applying contrastive decoding with Top-K masking to revise uncertain predictions by referencing the approximated representation of pretrained knowledge. LGCD requires no additional training or access to the original pretraining data. Extensive experiments with LGCD on multilingual multiple-choice and long-form QA tasks across nine languages demonstrate its strong effectiveness in mitigating hallucinations and enhancing factual accuracy in language-adapted models. Moreover, these results indicate that pretrained knowledge can be strategically reintroduced during decoding to promote factual multilingual generation.

## 1 Introduction

Large Language Models (LLMs) have demonstrated remarkable capabilities across a wide range of natural language tasks. A common practice to enhance their performance for specific languages or domains involves continual pretraining (CPT) or instruction fine-tuning (Gururangan et al., 2020; Zhang et al., 2024; Huang et al., 2023). While these adaptation techniques often inject new knowledge and improve task-specific abilities in the target language or domain, they frequently suffer from a critical drawback: *catastrophic forgetting* (Luo et al., 2023; OLMo et al., 2024; Li & Lee, 2024; Li et al., 2024; Kalajdzievski, 2024). This phenomenon leads to the degradation of general knowledge acquired during the initial pretraining phase, often resulting in increased factual inaccuracies or hallucinations (Ji et al., 2023; Luo et al., 2023; Li & Lee, 2024). Empirical studies confirm that LLMs undergoing CPT or instruction tuning can lose previously learned knowledge, sometimes prioritizing stylistic alignment or fluency in the target language over the factual consistency inherent in the original model (Luo et al., 2023).

---

*Corresponding author.

Mitigating catastrophic forgetting during adaptation is challenging. Ideally, one would retrain the model using a mixture of the original pretraining data and the new adaptation data. However, the original pretraining datasets for many state-of-the-art LLMs (e.g., LLaMA, Qwen) are generally undisclosed and inaccessible, though efforts towards fully open models like OLMo exist (OLMo et al., 2024). Furthermore, retraining from scratch or even extensive CPT demands prohibitive computational resources and time. Although various techniques aim to reduce forgetting during the training process (Gu et al., 2024; Huang et al., 2024; He et al., 2024; Wang et al., 2023b; Vo et al., 2024), they remain limited in preserving general knowledge, especially when adapting to new domains or languages. This limitation motivates the exploration of alternative approaches that can enhance the factuality of adapted LLMs without requiring further training or access to original pretraining data.

Recent research has highlighted the role of Feed-Forward Network (FFN) layers within the Transformer architecture as key-value memories, crucial for storing factual knowledge acquired during pretraining (Geva et al., 2020; Qiu et al., 2024; Dai et al., 2023). Inspired by this understanding, we hypothesize that the knowledge implicitly stored within the FFN weights of the original pretrained model can be explicitly leveraged to support the generation process of an adapted (e.g., continually pretrained or instruction-tuned) model at inference time, thereby improving its factual accuracy.

In this work, we propose LoRA-Gated Contrastive Decoding (LGCD), a novel training-free decoding method designed to enhance the factuality of LLMs, particularly those adapted for specific languages or domains. LGCD addresses the inherent trade-off between domain-specific fluency and general factual knowledge by dynamically switching between decoding strategies based on token-level confidence and applying contrastive decoding when necessary.

The framework of LGCD is characterized by three key components: First, it performs LoRA-based factual knowledge extraction from FFN layers and obtains a lightweight approximation of the pretrained model (PTM), by computing parameter differences between pretrained and adapted models and decomposing them using Singular Value Decomposition (SVD) to recover factual knowledge in FFN layers without modifying the language-adapted model (LAM). Second, it employs confidence-based dynamic gating that measures token-level confidence from the LAM and determines when to trigger factual knowledge injection, ensuring that domain fluency is preserved when the model is confident while leveraging pretrained knowledge when uncertainty arises. Third, it implements contrastive decoding with Top-K masking, which computes contrastive logits by subtracting the LAM's logits from the logits of the approximated PTM (aPTM), and applies this correction only to the top-K candidates predicted by the LAM. This selective adjustment injects factual knowledge while minimizing disruption to fluent generation.

We conduct a comprehensive evaluation of LGCD across nine diverse languages, highlighting its broad applicability in multilingual settings. Our experiments demonstrate LGCD's effectiveness across multiple evaluation settings, including multilingual multiple-choice benchmarks such as Global MMLU (Singh et al., 2024a) and multilingual TruthfulQA (Dac Lai et al., 2023) for domain-specific and general factual knowledge, long-form generation benchmarks such as Multi-FAct (Shafayat et al., 2024) for factual consistency, and long-form medical QA tasks for precise knowledge grounding in high-stakes domains.

Our contributions are threefold:

1. We propose LGCD, a novel training-free, decoding-time framework to mitigate hallucination and enhance factuality in language-adapted LLMs by leveraging knowledge from the original pretrained model through dynamic model switching and contrastive decoding.

2. We introduce specific techniques within LGCD, including LoRA-based knowledge extraction from FFN layers, confidence-based dynamic gating for token-level decision making, and contrastive decoding with Top-K masking.

3. We provide extensive empirical evidence demonstrating LGCD's effectiveness across multilingual multiple-choice QA and long-form generation tasks, using nine languages and twelve models. Our approach consistently outperforms adapted models without requiring additional training or external resources.

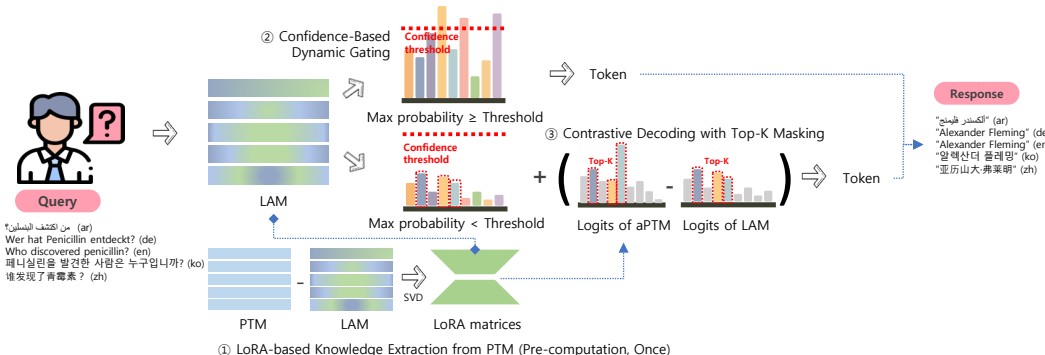

Figure 1: Overview of the LGCD framework.

## 2 RELATED WORK

### 2.1 HALLUCINATION MITIGATION IN LLM

Addressing hallucinations in LLMs involves various strategies, including improvements in training data and model architecture, fact-checking mechanisms, and integrating external knowledge sources like retrieval systems or knowledge graphs (Izacard & Grave, 2020; Wang et al., 2023a). While effective, the aforementioned external methods often introduce complexity or dependencies. Our proposed method, LGCD, focuses on an internal, decoding-time approach to mitigate hallucination without requiring external models or significant architectural changes.

### 2.2 DECODING STRATEGIES FOR FACTUALITY

Factual consistency in LLM decoding is often improved by tweaking output probabilities. For instance, Contrastive search (Su et al., 2022) combats repetition by picking tokens that are both likely and semantically distinct from prior context. Similarly, DoLa (Chuang et al., 2024) leverages deeper layers by contrasting logits across different internal layers of the same model. These methods reshape probability distributions based on disagreements or differences within the model or its variants. Our LGCD takes a different approach. Instead of merely contrasting probabilities, LGCD explicitly extracts and integrates factual knowledge from FFN layers of the original PTM–a process that is performed only once. This knowledge then directly influences the LAM's logits via a confidence-gated mechanism, offering a training-free solution to inject specific factual signals and enhance accuracy in adapted LLMs without further access to the PTM during inference.

### 2.3 KNOWLEDGE IN FEED-FORWARD NETWORKS

Previous research has shown that FFN layers within transformer models serve as key repositories for factual and world knowledge, often interpreted as key-value memories (Geva et al., 2020; Qiu et al., 2024; Dai et al., 2023). This perspective suggests that structured knowledge is encoded within their weights. Our LGCD leverages this by explicitly recovering factual knowledge from FFN layers of the original PTM. This is vital because catastrophic forgetting can degrade such knowledge in LAMs. LGCD achieves this through LoRA-based factual knowledge extraction: it calculates parameter differences between the PTM and LAM, then uses SVD to pinpoint and recover PTM's factual knowledge without altering the LAM. This precisely extracted knowledge is then dynamically injected during decoding, central to LGCD's factuality enhancement.

## 3 METHODOLOGY

This section introduces LGCD, a training-free decoding framework that combines the strengths of LAM and PTM. LGCD addresses the inherent trade-off between domain-specific fluency and general factual knowledge by dynamically switching between models based on token-level confidence and applying contrastive decoding when necessary. Figure 1 illustrates our framework.

The core motivation behind LGCD is grounded in the observation that FFN layers in Transformer architectures act as key-value neural memories for factual knowledge (Geva et al., 2020; Dai et al., 2023; Qiu et al., 2024). During CPT or instruction fine-tuning, this knowledge can be degraded due to catastrophic forgetting. LGCD mitigates this by explicitly recovering factual knowledge from the pretrained model's FFN layers and dynamically injecting it during decoding.

The LGCD framework consists of three components: (1) LoRA-based factual knowledge extraction from FFN layers, (2) confidence-based dynamic gating, and (3) contrastive decoding with Top-K masking.

### 3.1 LoRA-based Knowledge Extraction from FFN Layers

To capture factual knowledge preserved in the pretrained model, LGCD performs LoRA extraction from all FFN layers. Specifically, for each FFN layer $\ell$, we compute the parameter difference between the pretrained model $M_{\text{PTM}}$ and the language-adapted model $M_{\text{LAM}}$:

$$\Delta W_\ell = W_\ell^{\text{PTM}} - W_\ell^{\text{LAM}} \tag{1}$$

We then apply SVD:

$$\Delta W_\ell = U_\ell \Sigma_\ell V_\ell^\top \tag{2}$$

LoRA matrices are constructed by retaining the top-$r$ singular components:

$$A_\ell = U_\ell[:, :r] \cdot \sqrt{\Sigma_\ell[:r]} \tag{3}$$

$$B_\ell = \sqrt{\Sigma_\ell[:r]} \cdot V_\ell^\top[:r, :] \tag{4}$$

The pretrained FFN weight is approximated as:

$$W_\ell^{\text{aPTM}} = W_\ell^{\text{LAM}} + A_\ell B_\ell \tag{5}$$

Notably, this process is performed only once in the entire framework and allows LGCD to retrieve factual knowledge from the PTM without modifying the LAM directly or incurring additional memory overhead from deploying separate models. It is applied only to the FFN layers, leaving all other components of the LAM unchanged. To empirically validate this design decision, we compare different layer-wise LoRA approximation strategies in Appendix A.3 and find that targeting only FFN layers yields the best performance.

### 3.2 Confidence-Based Dynamic Gating

At each decoding step $t$, LGCD first queries the LAM to compute logits $\mathbf{l}_t^{\text{LAM}}$. The token-level confidence is measured as the maximum probability over the vocabulary:

$$c_t = \max \left( \text{softmax}(\mathbf{l}_t^{\text{LAM}}) \right) \tag{6}$$

A fixed confidence threshold $\tau$ determines the decision. $\tau$ in our setting is determined based on language-specific data availability, as detailed in Appendix A.8.

- If $c_t \geq \tau \to$ Decode with LAM.
- If $c_t < \tau \to$ Contrastive decode with the aPTM.

This dynamic gating balances the fluency of the LAM with the factual reliability of the aPTM.

### 3.3 Contrastive Decoding with Top-K Masking and Layer-wise Contrast

When contrastive decoding is triggered, LGCD first computes logits of aPTM $\mathbf{l}_t^{\text{aPTM}}$ approximated with LoRA:

$$\mathbf{l}_t^{\text{aPTM}} = \mathbf{l}_t^{\text{LAM}} + \text{LoRA}(\Delta W_\ell, \mathbf{h}_t^{\text{LAM}}) \tag{7}$$

To prevent contrastive decoding from selecting tokens with low probabilities across both models, LGCD applies Top-K masking, considering only the $K$ most probable tokens from $\mathbf{l}_t^{\text{LAM}}$:

$$\mathcal{T}_K = \text{TopK}(\mathbf{l}_t^{\text{LAM}}, K) \tag{8}$$

The contrastive logits are computed as:

$$\mathbf{l}_t^{\text{contrast}}[i] = \begin{cases} \text{if } i \in \mathcal{T}_K: & \mathbf{l}_t^{\text{LAM}}[i] + \beta \cdot \left(\mathbf{l}_t^{\text{aPTM}}[i] \quad - \alpha \cdot \mathbf{l}_t^{\text{LAM}}[i]\right) \\ \text{otherwise:} & -\infty \end{cases} \tag{9}$$

where $\beta$ is a hyperparameter controlling the overall contrastive weighting and $\alpha \in [0,1]$ controls the degree of down-weighting applied to the LAM logits within the correction term.

This formulation goes beyond standard contrastive decoding by introducing a correction term $\left(\mathbf{l}_t^{\text{aPTM}}[i] - \alpha \cdot \mathbf{l}_t^{\text{LAM}}[i]\right)$ that prioritizes the aPTM's factual knowledge while gently penalizing potentially overconfident LAM predictions. By tuning $\alpha$, we modulate the LAM's influence without abruptly overriding it. Hyperparameter details are provided in Appendix A.4, and for completeness we include the full decoding algorithm and pseudocode in Appendix A.2.

## 4 EXPERIMENTAL SETUP

We evaluate our model on two task types—multiple-choice QA and long-form generation—across nine target languages: Chinese (zh), German (de), Portuguese (pt), Arabic (ar), Persian (fa), Japanese (ja), Korean (ko), Indonesian (id), and Swahili (sw). Unless otherwise noted, all experiments are conducted using 12 models.

### 4.1 MULTIPLE-CHOICE QA

**Global MMLU.** To assess multilingual factual understanding, we use Global MMLU (Singh et al., 2024a), a culturally-aware extension of MMLU with 14K curated questions spanning 57 subjects in 42 languages. We report accuracy per language in zero- and five-shot settings for nine target languages.

**Multilingual TruthfulQA.** We use the MC1 version of TruthfulQA (Dac Lai et al., 2023; Lin et al., 2021a) across 31 languages. The number of questions per language varies, with most languages containing at least 700 items. We evaluate six models in zero-shot and five-shot settings.

### 4.2 LONG-FORM GENERATION

**Medical QA.** For high-stakes generative evaluation, we use multilingual medical QA datasets with expert-validated answers (Appendix A.5). Evaluation includes:

- **LLM-as-a-Judge:** GPT-4o performs pairwise comparisons with baselines (Li et al., 2023a).
- **Human Evaluation:** For 12 LAMs, we sample 20 questions each (240 total). Outputs are translated to English and rated by three experts on fluency, coherence, specificity, and factuality. Final labels (*Win/Tie/Lose*) are based on majority vote.

**Multi-FAct.** We evaluate factuality with Multi-FAct (Shafayat et al., 2024), which uses FActScore (Min et al., 2023) to decompose generations into atomic facts and verify them against trusted sources in multilingual settings.

### 4.3 MODELS & BASELINES

**Models.** We evaluate 12 publicly available LAMs from Hugging Face, each specialized for one of the 9 target languages (*zh, de, pt, ar, fa, ja, ko, id, sw*). All models are based on multilingual LLM backbones (mainly LLaMA-3 variants) and further adapted via CPT, instruction tuning, or both. Model selection was guided by language specificity, public availability, and community engagement (e.g., download count, active maintenance). Table 1 summarizes model specifications.

| Lang. | Model | CPT | Instr. tuning |
|---|---|---|---|
| zh | shenzhi-wang/Llama3-8B-Chinese-Chat | ✓ | ✓ |
| zh | hfl/llama-3-chinese-8b-instruct | ✓ | ✓ |
| de | DiscoResearch/Llama3-DiscoLeo-Instruct-8B-v0.1 | ✓ | ✓ |
| pt | rhaymison/gemma-portuguese-luana-2b | ✗ | ✓ |
| ar | MohamedRashad/Arabic-Orpo-Llama-3-8B-Instruct | ✗ | ✓ |
| fa | PartAI/Dorna-Llama3-8B-Instruct | ✗ | ✓ |
| ja | tokyotech-llm/Llama-3-Swallow-8B-Instruct-v0.1 | ✓ | ✓ |
| ja | elyza/Llama-3-ELYZA-JP-8B | ✓ | ✓ |
| ko | KISTI-KONI/KONI-Llama3-8B-Instruct-20240729 | ✓ | ✓ |
| ko | MLP-KTLim/llama-3-Korean-Bllossom-8B | ✓ | ✓ |
| id | GoToCompany/llama-3-8b-cpt-sahabatai-v1-instruct | ✓ | ✓ |
| sw | Jacaranda/UlizaLlama3 | ✓ | ✓ |

Table 1: LAMs used in our experiments. "✓" indicates application of CPT and/or instruction tuning.

**Baselines.** We compare LGCD against Nucleus Sampling (NS), DoLa (Chuang et al., 2024), TIES (Yadav et al., 2023a), and SLERP (Shoemake, 1985)[1]. This set covers widely used decoding methods and model-merging approaches, enabling a comprehensive assessment of LGCD.

| Lang. | Model | 0-shot | | | | | | 5-shot | | | | | |
|---|---|---|---|---|---|---|---|---|---|---|---|---|---|
| | | PTM | LAM | DoLa | TIES | SLERP | LGCD | PTM | LAM | DoLa | TIES | SLERP | LGCD |
| zh | hfl/llama-3-chinese-8b-instruct | 0.494 | 0.466 | 0.467 | 0.502 | 0.494 | **0.519** | 0.532 | 0.515 | 0.514 | 0.536 | 0.532 | **0.543** |
| zh | shenzhi-wang/Llama-3-8B-Chinese-Chat | 0.494 | 0.500 | 0.500 | 0.498 | 0.494 | **0.502** | 0.532 | **0.543** | 0.542 | 0.538 | 0.532 | 0.543 |
| de | DiscoResearch/Llama3-DiscoLeo-Instruct-8B-v0.1 | 0.514 | 0.486 | 0.492 | 0.540 | 0.514 | **0.546** | 0.558 | 0.548 | 0.550 | 0.566 | **0.584** | 0.574 |
| pt | rhaymison/gemma-portuguese-luana-2b | 0.353 | 0.316 | 0.278 | **0.357** | 0.353 | **0.357** | 0.325 | 0.316 | 0.298 | **0.330** | 0.325 | 0.324 |
| ar | MohamedRashad/Arabic-Orpo-Llama-3-8B-Instruct | 0.425 | 0.430 | 0.427 | 0.425 | 0.441 | **0.465** | 0.467 | 0.471 | 0.473 | 0.467 | **0.483** | 0.481 |
| fa | PartAI/Dorna-Llama3-8B-Instruct | **0.424** | 0.423 | 0.424 | 0.423 | 0.424 | 0.423 | 0.465 | 0.466 | **0.468** | 0.466 | 0.465 | 0.466 |
| ja | tokyotech-llm/Llama-3-Swallow-8B-Instruct-v0.1 | 0.481 | 0.478 | 0.479 | 0.462 | 0.456 | **0.507** | 0.481 | **0.527** | 0.527 | 0.525 | 0.481 | 0.525 |
| ja | elyza/Llama-3-ELYZA-JP-8B | 0.481 | 0.473 | 0.473 | 0.466 | 0.456 | **0.509** | 0.481 | 0.503 | 0.503 | 0.513 | 0.481 | **0.528** |
| ko | KISTI-KONI/KONI-Llama3-8B-Instruct-20240729 | 0.437 | 0.445 | 0.459 | 0.445 | 0.437 | **0.490** | 0.481 | 0.495 | 0.495 | 0.509 | 0.481 | **0.511** |
| ko | MLP-KTLim/llama-3-Korean-Bllossom-8B | 0.437 | 0.376 | 0.378 | 0.435 | 0.437 | **0.479** | 0.481 | 0.447 | 0.448 | 0.485 | 0.481 | **0.501** |
| id | GoToCompany/llama3-8b-cpt-sahabatai-v1-instruct | 0.476 | 0.530 | **0.531** | 0.486 | 0.476 | 0.527 | 0.529 | 0.576 | **0.577** | 0.530 | 0.568 | 0.566 |
| sw | Jacaranda/UlizaLlama3 | 0.359 | 0.362 | 0.363 | 0.355 | 0.359 | **0.399** | 0.388 | 0.367 | 0.372 | 0.390 | **0.432** | 0.409 |
| | Average | 0.448 | 0.441 | 0.439 | 0.449 | 0.445 | **0.477** | 0.477 | 0.481 | 0.481 | 0.488 | 0.487 | **0.498** |

Table 2: Evaluation accuracy of 12 models on Global MMLU benchmark under 0-shot and 5-shot settings using various decoding and merging strategies. PTM refers to the performance of the pretrained Model, while LAM denotes the language-adapted model, both evaluated using Nucleus Sampling (NS). DoLa represents the results when applying the DoLa decoding strategy to the LAM.

## 5 EXPERIMENTAL RESULTS

We evaluate LGCD on two multilingual multiple-choice QA benchmarks—Global MMLU and Multilingual TruthfulQA—under zero-shot and five-shot settings. Comparisons include decoding-time baselines (DoLa), model merging methods (TIES, SLERP), and standard nucleus sampling (NS) for both pretrained (PTM) and language-adapted models (LAM).

### 5.1 GLOBAL MMLU

Table 2 shows accuracy across 12 LAMs covering 9 languages. LGCD achieves the best average performance in both zero-shot and five-shot settings, outperforming all decoding and merging baselines.

In the zero-shot setting, relative to the LAM, LGCD improves accuracy in 10 of 12 cases. Gains are largest where the LAM lags the original PTM (Korean: +4.5–10.3 pp; German: +6.0 pp; Japanese: +2.9–3.6 pp; Portuguese: +4.1 pp), indicating that LGCD effectively recovers knowledge lost during adaptation.

In contrast, in the five-shot setting, LGCD continues to provide consistent improvements, especially in high-resource languages such as Chinese and German. This indicates that LGCD not only recovers forgotten knowledge but also scales robustly with richer context, effectively balancing domain adaptation and pretrained factuality across diverse conditions.

Overall, LGCD consistently improves multilingual QA performance, showing robustness across culturally diverse and knowledge-heavy questions where language adaptation typically disrupts general knowledge.

### 5.2 MULTILINGUAL TRUTHFULQA

Results on Multilingual TruthfulQA (Table 3) further validate LGCD's ability to enhance factuality. In the zero-shot setting, LGCD achieves the highest average accuracy, outperforming both decoding and merging baselines across all evaluated languages.

LGCD demonstrates consistent improvements across all tested languages, with particularly notable gains in Portuguese (+10.8 pp), Arabic (+9.2 pp), and German (+3.9 pp). These improvements hold in the five-shot setting as well, where LGCD shows even larger gains, reinforcing the method's robustness across different prompting regimes.

---

[1]https://github.com/Digitous/LLM-SLERP-Merge

| | | 0-shot | | | | | | 5-shot | | | | | |
|---|---|---|---|---|---|---|---|---|---|---|---|---|---|
| Lang. | Model | PTM | LAM | DoLa | TIES | SLERP | LGCD | PTM | LAM | DoLa | TIES | SLERP | LGCD |
| zh | hfl/llama-3-chinese-8b-instruct | 0.349 | **0.353** | 0.312 | 0.352 | 0.335 | 0.352 | 0.379 | 0.390 | 0.392 | 0.381 | 0.363 | **0.471** |
| zh | shenzhi-wang/Llama3-8B-Chinese-Chat | 0.349 | **0.363** | 0.326 | 0.357 | **0.363** | 0.357 | 0.379 | 0.391 | 0.360 | 0.387 | 0.400 | **0.484** |
| de | DiscoResearch/Llama3-DiscoLeo-Instruct-8B-v0.1 | 0.317 | 0.343 | 0.320 | 0.325 | 0.330 | **0.382** | 0.367 | 0.367 | 0.354 | 0.373 | **0.393** | 0.391 |
| pt | rhaymison/gemma-portuguese-luana-2b | 0.272 | 0.301 | 0.268 | 0.283 | 0.302 | **0.409** | 0.319 | 0.325 | 0.299 | 0.322 | 0.321 | **0.431** |
| ar | MohamedRashad/Arabic-Orpo-Llama-3-8B-Instruct | 0.325 | 0.308 | 0.331 | 0.326 | 0.318 | **0.400** | 0.376 | 0.360 | 0.323 | 0.383 | 0.365 | **0.426** |
| id | GoToCompany/llama3-8b-cpt-sahabatai-v1-instruct | 0.329 | 0.334 | 0.334 | 0.341 | 0.334 | **0.353** | 0.378 | 0.370 | 0.369 | 0.373 | 0.365 | **0.406** |
| | Average | 0.323 | 0.334 | 0.315 | 0.331 | 0.330 | **0.376** | 0.366 | 0.367 | 0.349 | 0.370 | 0.368 | **0.435** |

Table 3: Evaluation accuracy of 6 models on multilingual TruthfulQA benchmark under 0-shot and 5-shot settings using different merging and decoding strategies.

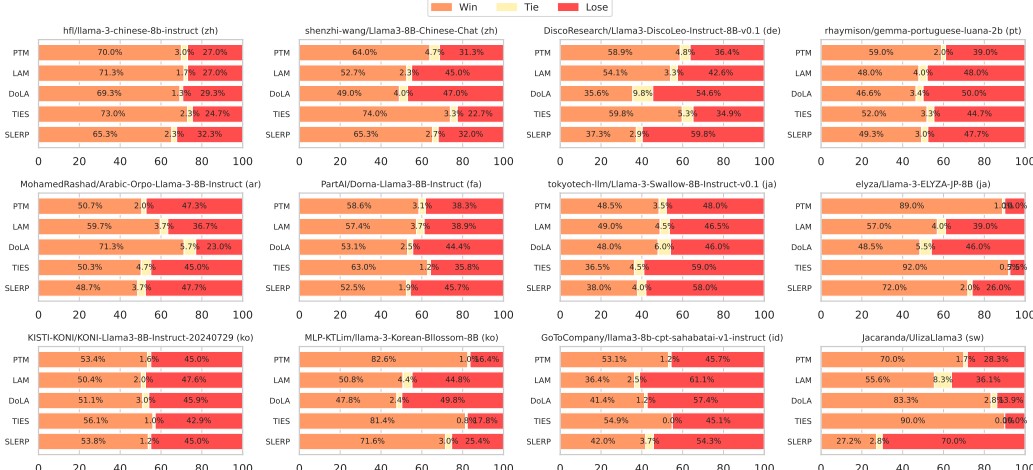

Figure 2: Pairwise comparison results of our model vs. baselines (evaluated by GPT-4o)

These results confirm that LGCD effectively resists plausible but incorrect generations, excelling in settings in TruthfulQA that probe a model's ability to distinguish truth from commonly held misconceptions.

## 5.3 LONG-FORM MEDICAL QA

We further evaluate LGCD in high-stakes, long-form medical question answering tasks across 12 LAMs. Each model is tested in its target language using domain-specific medical QA datasets. Evaluation is conducted via two complementary methods: (1) GPT-4o as an automatic judge performing pairwise comparisons, and (2) expert human preference evaluations on fluency, coherence, and factual correctness.

**LLM-as-a-Judge Evaluation.** Figure 2 shows GPT-4o-based preference comparisons. LGCD achieves the highest win rates in most languages, outperforming PTM, LAM, and all decoding or merging baselines. On average, LGCD is preferred over PTM in 63.1% of cases, over LAM in 53.5%, and over DoLa (53.8%) and SLERP (51.9%), while performing competitively with TIES (65.3%).

**Human Preference Evaluation.** We further assess LGCD through human preference evaluation. As shown in Figure 3, LGCD is consistently favored across baselines, achieving higher win rates than PTM (52.0%), DoLA (45.8%), and SLERP (45.8%). These results reinforce LGCD's ability to generate fluent and factually grounded answers in complex medical domains. Despite cross-lingual pooling, the consistent preference trend across models suggests that LGCD offers a robust and training-free alternative to improve factuality in long-form, domain-specific generation.

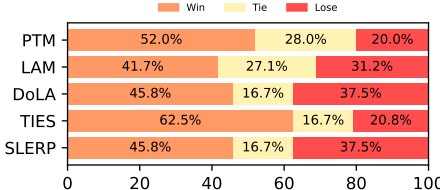

Figure 3: Human preference comparison between our model and baseline models

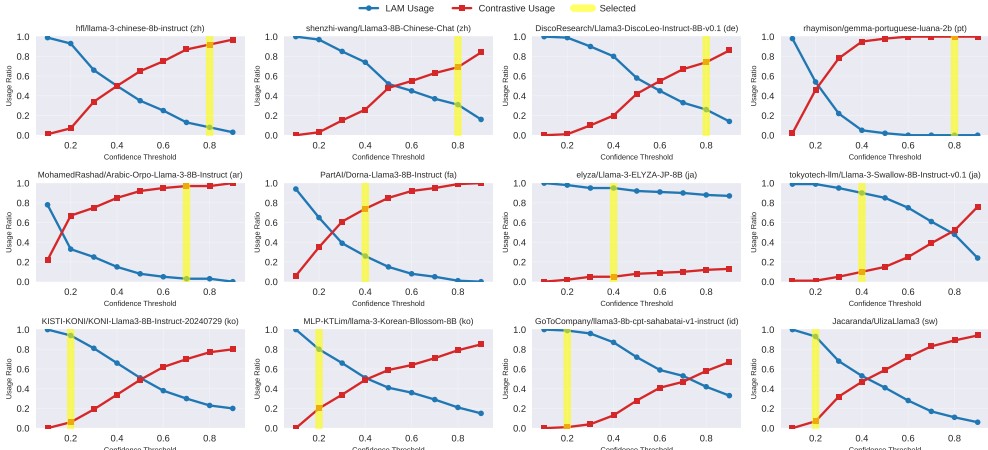

Figure 4: Contrastive decoding behavior in LGCD across 12 LAMs. For each model, we plot the token-level usage ratio of the base LAM (blue) and contrastive candidate (red) as a function of confidence threshold $\tau$. The yellow vertical line marks the threshold selected for that model. LGCD dynamically adjusts model usage based on token uncertainty, striking a balance between domain alignment and factual recall.

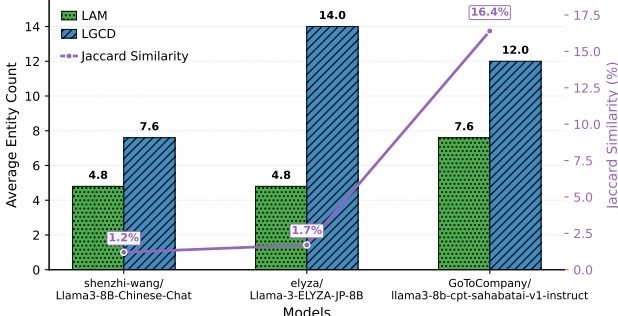

Figure 5: Average entity count (left) and Jaccard similarity (right) between LGCD and LAM outputs for Chinese, Japanese, and Indonesian models

**Contrastive Usage Analysis** Figure 4 shows the proportion of tokens decoded by the LAM (blue) and via contrastive decoding (red) as the confidence threshold $\tau$ varies. The yellow band indicates the threshold selected for each model.

In higher-resource languages (e.g., zh, de, pt, ar, fa), models adopt higher thresholds (typically $\tau = 0.7$–$0.8$), leading to a sharp increase in contrastive usage. This reflects the stronger performance of the pretrained model in these languages, allowing LGCD to revise uncertain predictions more effectively.

In contrast, for lower-resource languages (e.g., ja, ko, id, sw), LAM usage remains high even at high thresholds. These models tend to produce overconfident outputs, possibly due to limited token coverage during adaptation. Since the PTM is also less reliable in these settings, LGCD uses lower thresholds to favor the LAM, which yields better results despite overconfidence.

To probe where LGCD's gains come from in long-form generation, we analyze named-entity behavior under a general Named Entity Recognition (NER) schema and compare LGCD to the underlying LAM. Further details are provided in the Appendix A.7. Figure 5 summarizes two signals: (i) average number of entities extracted per output and (ii) Jaccard overlap between the entity sets from LGCD and the LAM. Across three representative LAMs (Chinese, Japanese, Indonesian), LGCD consistently produces more entities than the LAM while the set overlap remains low–moderate ($\approx 1$–$16\%$). This pattern suggests LGCD is not merely echoing the LAM's choices but is adding complementary, likely factual, mentions that the LAM omits. Linking back to the Contrastive Usage

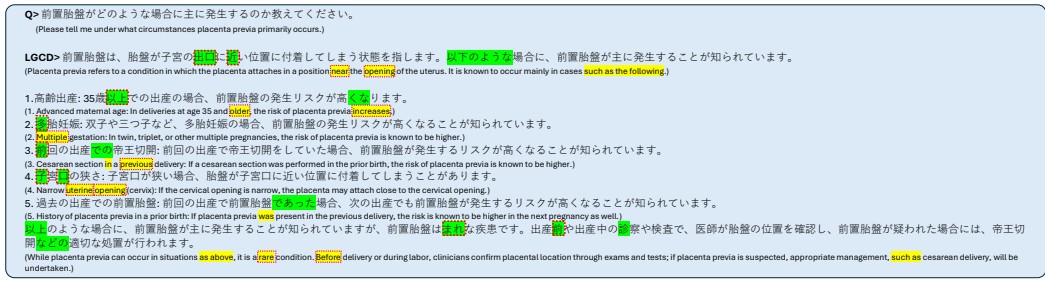

Figure 6: Token-level view on a Japanese medical QA example (elyza/Llama-3-ELYZA-JP-8B). LGCD output in Japanese (top) with English alignment (bottom). Tokens highlighted in green mark time-steps where LGCD switched on contrastive decoding; their aligned tokens in the translation are highlighted in yellow. Tokens outlined in red indicate decisive positions that steered the continuation toward the factual answer.

| Lang. | Model | LAM | LGCD | Δ |
|---|---|---|---|---|
| zh | hfl/llama-3-chinese-8b-instruct | 0.260 | 0.246 | −0.014 |
| zh | shenzhi-wang/Llama3-8B-Chinese-Chat | 0.229 | **0.313** | +0.084 |
| de | DiscoResearch/Llama3-DiscoLeo-Instruct-8B-v0.1 | 0.387 | **0.520** | +0.133 |
| pt | rhaymison/gemma-portuguese-luana-2b | **0.288** | 0.267 | −0.021 |
| ja | tokyotech-llm/Llama-3-Swallow-8B-Instruct-v0.1 | **0.267** | 0.197 | −0.070 |
| ja | elyza/Llama-3-ELYZA-JP-8B | 0.298 | **0.302** | +0.004 |
| ko | KISTI-KONI/KONI-Llama3-8B-Instruct-20240729 | 0.196 | **0.218** | +0.022 |
| ko | MLP-KTLim/llama-3-Korean-Bllossom-8B | 0.189 | **0.196** | +0.007 |
| id | GoToCompany/llama3-8b-cpt-sahabatai-v1-instruct | 0.334 | **0.547** | +0.212 |
| | **Average** | 0.272 | **0.312** | **+0.040** |

Table 4: Comparison of Multi-FAct score between LAM and LGCD. Models with a LAM score below 0.05 were excluded, resulting in 9 evaluated models.

Analysis, these results explain why factuality can improve even when overall contrastive usage is low—e.g., in Japanese (elyza/Llama-3-ELYZA-JP-8B) and Indonesian (GoToCompany/llama3-8b-cpt-sahabatai-v1-instruct).

Figure 6 further illustrates that LGCD intervenes only sparsely, yet the activated gates coincide with tokens that are not only entity mentions but also those carrying decisive factual content. Despite their small number, these targeted interventions are sufficient to steer the generation toward factual answers. A direct comparison of LAM and LGCD outputs for the Japanese medical QA example is provided in the Appendix A.10.

### 5.4 MULTI-FACT

On the Multi-FAct benchmark, LGCD improves factual consistency in 6 of 9 models, with a +0.04 average gain over LAMs (Table 4). Notable gains in Indonesian, German and Korean confirm LGCD's effectiveness in enhancing factuality for long-form generation.

### 5.5 THROUGHPUT

Table 5 presents the decoding speed of different generation strategies measured on a single A100 GPU. Results are averaged over responses to 100 questions using the `hfl/llama-3-chinese-8b-instruct` model. Among baselines, greedy search achieves the highest throughput (19.21 tokens/sec), followed by nucleus sampling (17.47). Contrastive Search is slower (11.87) due to the similarity penalty over top-k candidates, though it runs in a single forward pass without reranking. LGCD slows decoding by querying the aPTM when the LAM lacks confidence. However, the overhead varies with the confidence

| Decoding Strategy | Throughput (Token/s) |
|---|---|
| Greedy search | 19.21 |
| Nucleus sampling | 17.47 |
| Contrastive search | 11.87 |
| DoLa | 16.81 |
| LGCD-0.2 | 14.37 |
| LGCD-0.4 | 10.32 |
| LGCD-0.6 | 10.32 |
| LGCD-0.8 | 10.22 |

Table 5: Average decoding throughput for each strategy. For LGCD-$\tau$, the numeric suffix denotes the confidence threshold $\tau$ used in decoding.

threshold $\tau$: lower thresholds (e.g., LGCD-0.2) lead to fewer contrastive decisions and thus higher throughput (14.37), while higher $\tau$ (e.g., LGCD-0.8) leads to more aggressive factual intervention and slower speed (10.22). Notably, LGCD-0.2 approaches the speed of DoLa (16.81), demonstrating that factual calibration can be achieved without substantial efficiency loss when appropriately tuned. This tunability enables LGCD to balance quality and speed for practical deployment.

## 6 LIMITATION

While LGCD effectively enhances factuality in language-adapted models, several limitations remain. First, our framework relies on the availability of the original PTM weights for a one-time offline knowledge extraction process. Although the PTM is not required during inference , this initial requirement may limit the applicability of LGCD in scenarios where PTM weights are entirely inaccessible or undisclosed. Second, the effectiveness of our dynamic gating is tied to the reliability of logit-based confidence signals. If the model is extremely overconfident or if logit-derived uncertainty is miscalibrated. Future work could explore more robust uncertainty measures to further enhance the reliability of the gating process across diverse model architectures.

## 7 CONCLUSION

We present LGCD, a novel training-free method that enhances the factuality of language-adapted LLMs by dynamically injecting pretrained knowledge during inference. By extracting factual signals from FFN layers via LoRA-based decomposition and applying contrastive decoding conditioned on token-level confidence, LGCD effectively mitigates catastrophic forgetting without retraining or access to pretraining data. Extensive multilingual experiments across multiple-choice QA and long-form generation tasks demonstrate that LGCD consistently improves factual accuracy over both decoding and model-merging baselines, offering a practical, scalable solution for factuality preservation in domain-specialized LLMs.

## 8 ACKNOWLEDGMENTS

This research was supported by the Korea Institute of Science and Technology Information (KISTI) in 2026 (No. (KISTI)K26L3M1C1), aimed at developing KONI (KISTI Open Neural Intelligence), a large language model specialized in science and technology.

Additionally, this work was also supported by the National Research Foundation of Korea (NRF) grant funded by the Korea government (MSIT) (No.RS-2022-NR068758)

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

## A  APPENDIX

### A.1  LLM USAGE DISCLOSURE

We used LLMs in the following limited ways to aid our research process:

- **Writing Assistance**: ChatGPT-5 was used to polish and refine writing, including grammar correction, translation for clarity, and LaTeX table formatting support. All generated or suggested content was carefully reviewed and edited by the authors before inclusion.
- **Evaluation (LLM-as-Judge)**: GPT-4o was employed as an automatic evaluator for long-form medical QA performance, and internally within the Multi-FAct benchmark, following established "LLM-as-judge" protocols.
- **Code Debugging**: ChatGPT-5 was used in a supporting role to debug code. The authors independently verified the correctness of all outputs.

The LLMs did not contribute to research ideation or the design of experiments. All final content and claims in the paper remain the responsibility of the authors.

## A.2 Decoding Process

Algorithm 1 summarizes the complete LGCD procedure. At each timestep $t$, the framework evaluates token-level confidence $c_t$ from the LAM. When $c_t \geq \tau$, decoding proceeds with the LAM alone using Top-K sampling. When $c_t < \tau$, indicating potential factual uncertainty, LGCD activates the contrastive mechanism: (1) computing LoRA-approximated pretrained logits, (2) dynamically gating based on token-level confidence, and (3) applying contrastive weighting within the Top-K space. This confidence-driven approach enables LGCD to dynamically balance domain fluency and factual accuracy without requiring model retraining, making it particularly suitable for scenarios where language adaptation may compromise general knowledge.

---

**Algorithm 1** LoRA-Gated Contrastive Decoding (LGCD)

---

**Require:** Language-Adapted Model $M_{\text{LAM}}$, Approximated Pretrained Model $M_{\text{aPTM}}$ (for $\Delta W_\ell$), confidence threshold $\tau$, Top-K size $K$, contrastive weighting hyperparameter $\beta$
**Ensure:** Generated output sequence $\mathcal{Y}$
1: Initialize $\mathcal{Y} = [\ ]$
2: **while** $\mathcal{Y}[t] \neq \text{<eos>}, t = 1, 2, \ldots$ **do**
3:     Compute LAM logits $\mathbf{l}_t^{\text{LAM}}$
4:     Compute token-level confidence $c_t$:
5:        $c_t = \max(\text{softmax}(\mathbf{l}_t^{\text{LAM}}))$
6:     **if** $c_t \geq \tau$ **then**
7:        Apply Top-K masking: $\mathcal{T}_K = \text{TopK}(\mathbf{l}_t^{\text{LAM}}, K)$
8:        Select next token $y_t$ by sampling from the masked $\mathbf{l}_t^{\text{LAM}}$
9:     **else**
10:        Compute LoRA-approximated pretrained logits:
11:          $\mathbf{l}_t^{\text{aPTM}} = \mathbf{l}_t^{\text{LAM}} + \text{LoRA}(\Delta W_\ell, \mathbf{h}_t^{\text{LAM}})$
12:        Apply Top-K masking: $\mathcal{T}_K = \text{TopK}(\mathbf{l}_t^{\text{LAM}}, K)$
13:        Initialize $\mathbf{l}_t^{\text{contrast}} = [-\infty, \ldots, -\infty] \in \mathbb{R}^{\text{vocab\_size}}$
14:        **for** each token $i$ in vocabulary **do**
15:          **if** $i \in \mathcal{T}_K$ **then**
16:            Compute contrastive logit:
17:             $\mathbf{l}_t^{\text{contrast}}[i] = \mathbf{l}_t^{\text{LAM}}[i] + \beta \cdot \left(\mathbf{l}_t^{\text{aPTM}}[i] - \alpha \cdot \mathbf{l}_t^{\text{LAM}}[i]\right)$
18:          **else**
19:            $\mathbf{l}_t^{\text{contrast}}[i] = -\infty$
20:          **end if**
21:        **end for**
22:        Select next token $y_t$ by sampling from $\mathbf{l}_t^{\text{contrast}}$
23:     **end if**
24:     Append $y_t$ to output sequence $\mathcal{Y}$
25: **end while**
26: **return** $\mathcal{Y}$

---

## A.3 LoRA-Based PTM Approximation: FFN vs QV vs All-Layer Comparison

LGCD relies on approximating the PTM by applying LoRA-based updates to the LAM, using low-rank matrices derived from the difference between PTM and LAM parameters. In our primary design, this approximation targets only the FFN layers, based on prior research showing that FFNs encode core factual knowledge in LLMs (Geva et al., 2020; Qiu et al., 2024; Dai et al., 2023).

To assess the importance of this design choice, we compare three strategies for selecting which layers to approximate via LoRA during the offline distillation step:

- **QV-only:** Apply LoRA decomposition only to the attention projection matrices (Q, V).

- **FFN-only:** Apply LoRA only to FFN layers (our default).

- **All Layers:** Apply LoRA to both FFN and attention projection layers.

Table 6 shows the performance of LGCD using each variant on the Global MMLU benchmark under the zero-shot setting. Results show that FFN-only approximation consistently achieves the best or comparable performance, while QV-only performs worse on nearly all models. Approximating all layers introduces marginal gains in a few cases but often results in unstable or degraded performance, suggesting that unnecessary modification of attention layers may introduce noise. These findings empirically validate our FFN-only design as both effective and efficient for factuality-oriented approximation of PTMs.

| Lang. | Model | QV-only | FFN-only | All Layers |
|---|---|---|---|---|
| zh | hfl/llama-3-chinese-8b-instruct | 0.485 | 0.519 | **0.520** |
| zh | shenzhi-wang/Llama3-8B-Chinese-Chat | 0.500 | 0.502 | **0.503** |
| de | DiscoResearch/Llama3-DiscoLeo-Instruct-8B-v0.1 | 0.520 | **0.546** | 0.544 |
| pt | rhaymison/gemma-portuguese-luana-2b | 0.330 | **0.357** | 0.340 |
| ar | MohamedRashad/Arabic-Orpo-Llama-3-8B-Instruct | 0.460 | **0.465** | 0.460 |
| fa | PartAI/Dorna-Llama3-8B-Instruct | **0.423** | **0.423** | **0.423** |
| ja | tokyotech-llm/Llama-3-Swallow-8B-Instruct-v0.1 | 0.499 | **0.507** | 0.503 |
| ja | elyza/Llama-3-ELYZA-JP-8B | 0.499 | **0.509** | **0.509** |
| ko | KISTI-KONI/KONI-Llama-3-8B-Instruct-20240729 | 0.484 | **0.490** | 0.484 |
| ko | MLP-KTLim/llama-3-Korean-Bllossom-8B | 0.471 | **0.479** | 0.461 |
| id | GoToCompany/llama3-8b-cpt-sahabatai-v1-instruct | **0.533** | 0.527 | 0.528 |
| sw | Jacaranda/UlizaLlama3 | 0.375 | **0.399** | 0.395 |
| **Average** | | 0.465 | **0.477** | 0.472 |

Table 6: Ablation study on LoRA-based PTM approximation: comparing factual accuracy on Global MMLU (0-shot) when contrastive decoding uses knowledge reconstructed from different layer subsets.

## A.4 CONFIGURATIONS FOR LGCD

LGCD is configured with task-specific decoding strategies to address the differing demands of multiple-choice QA and long-form QA. While both settings use the same LoRA-based decomposition applied to all FFN layers with a fixed rank of 32, the confidence threshold and usage of contrastive decoding vary by task.

For **multiple-choice QA**, we use a fixed confidence threshold of $\tau = 0.9$ across all languages. This high threshold leads to frequent activation of contrastive decoding, ensuring that the model does not rely solely on the LAM but actively considers both the LAM and the PTM when scoring candidate answers. In this setup, LGCD functions not merely as a fallback for low-confidence predictions, but as a mechanism to integrate knowledge from both models during answer selection, enhancing factual reliability without requiring additional training. Decoding uses top-$k$ sampling ($k = 100$) with temperature 0.7 and top-$p = 0.9$, and all contrastive updates are scaled with $\beta = 1.0$.

For **long-form QA**, the decoding configuration remains identical in terms of sampling and contrastive scaling, but the confidence threshold $\tau$ is language-specific, calibrated according to resource availability (see Appendix A.8). This allows LGCD to adaptively determine when to apply contrastive decoding based on the reliability of the LAM for each language—more frequently for low-resource languages and conservatively for high-resource ones.

| Component | Multiple-Choice QA | Long-Form QA |
|---|---|---|
| Temperature | 0.7 | 0.7 |
| Top-$k$ | 100 | 100 |
| Top-$p$ | 0.9 | 0.9 |
| Contrastive $\beta$ | 1.0 | 1.0 |
| Contrastive $\alpha$ | 0.1 | 0.1 |
| LoRA (Rank / Scope) | 32 / All FFN layers | 32 / All FFN layers |
| Confidence $\tau$ | Fixed (0.9) | Language-specific |
| Contrastive Use | Joint scoring | Factual correction |

Table 7: LGCD decoding configurations for multiple-choice vs. long-form QA.

## A.5 Evaluation Datasets for Multilingual Medical Question Answering

This appendix provides a detailed account of the datasets employed to evaluate the multilingual medical question answering capabilities of the models discussed in this study. We have curated a collection of open-ended medical QA datasets across various languages, drawing from both established benchmarks and language-specific resources.

Our evaluation utilizes open-ended medical question answering data. For several languages, the Healthcare QA task from the AIR-Bench 24.05 benchmark (`https://github.com/AIR-Bench/AIR-Bench`) serves as the primary data source. This task comprises questions within the medical domain designed to assess a model's ability to provide informative responses. Specifically, the original Healthcare QA data from AIR-Bench was directly used for evaluation in Arabic and German.

For a broader linguistic evaluation where native high-quality long-form medical QA datasets were not readily available, we constructed evaluation sets by translating the English version of the Healthcare QA task from AIR-Bench. These translations were systematically performed using Google Translate. This methodology was applied to generate the evaluation datasets for Persian, Indonesian, Japanese, Portuguese, and Swahili. This approach enables a comparative analysis based on a consistent source structure across these languages, although with potential translation artifacts.

In addition to the AIR-Bench based datasets, we incorporated independent, language-specific medical QA datasets for Korean and Chinese:

- **Korean:** We used the `ChuGyouk/GenMedGPT-5k-ko` dataset (`https://huggingface.co/datasets/ChuGyouk/GenMedGPT-5k-ko`) for long-form medical QA in Korean. This dataset, containing approximately 5,000 question-answer pairs, is a Korean translation, performed using DeepL, of medical QA data sourced from `https://github.com/Kent0n-Li/ChatDoctor`.

- **Chinese:** For Chinese medical QA, we utilize the `cMedQA2` dataset (`https://github.com/zhangsheng93/cMedQA2`). This dataset is a dedicated resource for medical QA in Chinese, comprising a collection of questions and corresponding expert answers within the medical domain.

## A.6 Baseline Configuration Details

We compare LGCD against both decoding-based and model-merging baselines. All decoding strategies use the same chat template with `max_new_tokens = 2048`.

### Decoding-Based Baselines

We evaluate five decoding strategies: greedy decoding (GS), contrastive search (CS), nucleus sampling (NS), and DoLa. All hyperparameters follow either the original paper or Hugging Face implementation standards.

In particular, DoLa adopts a task-sensitive contrastive scheme: it uses higher transformer layers for multiple-choice QA and lower layers for long-form generation, following prior findings on factual calibration via depth selection.

| Method | Hyperparameters |
|---|---|
| GS | Hugging Face default (greedy decoding) |
| CS | `penalty_alpha = 0.6, top_k = 4` |
| NS | `do_sample = True, temperature = 0.7, top_p = 0.9` |
| DoLa | `do_sample = False` |
| | `dola_layers = "high"` (Multiple-Choice QA) |
| | `dola_layers = "low"` (Long-Form QA) |

Table 8: Decoding hyperparameter settings for baseline methods.

MODEL-MERGING BASELINES

We further compare LGCD with two weight-space integration baselines—**SLERP** and **TIES**—implemented using the `mergekit` framework. Both methods perform symmetric merging between the PTM and the LAM with the following common settings:

- Merge ratio: 0.5 (LAM) : 0.5 (PTM)

- Merge range: all transformer layers

- Data type: bfloat16

- Base model: LAM

- Other parameters: mergekit defaults

### A.7 NER-BASED FACTUALITY PROBE

**NER extraction.** We measure entity-level differences between LGCD and the baseline LAM by applying a general-purpose NER extractor to every generated output. Specifically, we use GPT-4o as the extractor. The schema is based on coarse-grained categories commonly used in NER (e.g., PERSON, ORGANIZATION, LOCATION, DATE, NUMBER), and we extended it with DISEASE to capture domain-relevant mentions. In total, we use 13 categories. Each entity is returned with its surface form, category, and character offset.

**Comparison metrics.** For each output pair, we form entity sets $E_{\text{LGCD}}$ and $E_{\text{LAM}}$. We compute Jaccard similarity $\frac{|E_{\text{LGCD}} \cap E_{\text{LAM}}|}{|E_{\text{LGCD}} \cup E_{\text{LAM}}|}$. We also track absolute entity counts per output. This setup allows us to test whether factuality gains from LGCD are reflected in increased coverage or correction of entity mentions beyond those produced by the base LAM.

### A.8 LANGUAGE-SPECIFIC CONFIDENCE THRESHOLDS BASED ON DATA AVAILABILITY

Our decoding framework dynamically balances contributions between a language-adapted model and a pretrained multilingual backbone based on a token-level confidence threshold. When the model's token-level confidence falls below the threshold, we incorporate contrastive logits from the pretrained model to supplement or correct uncertain predictions.

Given that the quality and extent of training for language-adapted models are often opaque, especially in multilingual settings, we estimate data availability using the OSCAR 22.01 corpus Caswell et al. (2021); Abadji et al. (2022); Abadji et al. (2021) as a public proxy for the quantity of available web-scale text per language. Languages are categorized into three tiers—High, Medium, and Low—based on word count.

We assign lower thresholds (e.g., 0.1) to low-resource languages such as Korean (ko), Indonesian (id), and Swahili (sw), allowing the pretrained model's generalized knowledge to exert greater influence. Conversely, for high-resource languages like German (de) and Chinese (zh), which have abundant data, we set a higher threshold (e.g., 0.8), favoring the language-specific model. Medium-resource languages such as Japanese (ja) and Persian (fa) are assigned intermediate values (e.g., 0.4). This approach ensures informed integration of pretrained knowledge proportional to resource availability.

| Category | Languages (ISO code) | Word Count |
|---|---|---|
| **High** ($\geq$ 2B) | German (de) | 46.8B |
| | Chinese (zh) | 23.1B |
| | Portuguese (pt) | 18.4B |
| **Medium** (0.5–2B) | Persian (fa) | 6.4B |
| | Arabic (ar) | 6.1B |
| | Japanese (ja) | 5.6B |
| **Low** ($<$ 0.5B) | Korean (ko) | 3.9B |
| | Indonesian (id) | 2.0B |
| | Swahili (sw) | $\approx$7MB |

Table 9: Language data categorization based on the public OSCAR 22.01 corpus.

## A.9 EVALUATION ON EXTREMELY LOW-RESOURCE LANGUAGES

To address concerns regarding the definition of "low-resource" languages and to test the robustness of LGCD, we conduct additional experiments on extremely low-resource languages as categorized by Nigatu et al. (2024). Specifically, we evaluate three South African languages: Zulu, Tswana, and Northern Sotho.

**Experimental Setup** Following our established evaluation protocol, we utilize `meta-llama/Llama-3.1-8B-Instruct` as the PTM and `africa-intelligence/llama-8b-south-africa` as the LAM. We conduct evaluations on two benchmarks:

- **Global MMLU**: Since standard Global MMLU benchmarks are not available for these languages, we translated the English portion of Global MMLU into Zulu, Tswana, and Northern Sotho using GPT-5. We report accuracy under both 0-shot and 5-shot settings, comparing LGCD against the PTM, LAM, DoLa, TIES, and SLERP baselines.

- **Healthcare QA**: We adapt the Healthcare QA task from the AIR-Bench 24.05 benchmark, translated into the target languages using GPT-5 following the methodology described in Appendix A.5. GPT-5 was employed as an LLM-as-a-judge to perform pairwise comparisons, scoring the outputs on factual accuracy and fluency. We compare LGCD against the LAM, DoLa, TIES, and SLERP.

Consistent with our strategy for low-resource tiers, we applied a confidence threshold of $\tau = 0.2$.

**Global MMLU Results** Table 10 and Table 11 present the accuracy on the translated Global MMLU benchmark under 0-shot and 5-shot settings, respectively. In the 0-shot setting, LGCD achieves the highest average accuracy (0.459), outperforming all baselines including DoLa (0.456). In the 5-shot setting, LGCD matches DoLa with the highest average accuracy of 0.432, while consistently outperforming TIES and SLERP. Notably, LGCD achieves the best performance on Tswana in both settings, demonstrating its robustness across different evaluation conditions.

| Language | PTM | LAM | DoLa | TIES | SLERP | LGCD |
|---|---|---|---|---|---|---|
| Zulu | 0.374 | 0.470 | **0.474** | 0.424 | 0.446 | 0.468 |
| Tswana | 0.358 | 0.448 | 0.452 | 0.386 | 0.446 | **0.456** |
| Northern Sotho | 0.366 | 0.438 | 0.442 | 0.438 | **0.460** | 0.454 |
| Avg | 0.366 | 0.452 | 0.456 | 0.416 | 0.451 | **0.459** |

Table 10: 0-shot accuracy on the translated Global MMLU benchmark for extremely low-resource languages.

**Healthcare QA Results** Tables 12–15 present the pairwise comparison results of LGCD against each baseline on the Healthcare QA task. Against the LAM (Table 12), LGCD achieves a clear

| Language | PTM | LAM | DoLa | TIES | SLERP | LGCD |
|---|---|---|---|---|---|---|
| Zulu | 0.392 | 0.430 | **0.434** | 0.414 | 0.422 | 0.431 |
| Tswana | 0.390 | 0.440 | 0.440 | 0.420 | 0.430 | **0.442** |
| Northern Sotho | 0.378 | 0.408 | 0.421 | 0.418 | 0.414 | **0.422** |
| Avg | 0.387 | 0.426 | **0.432** | 0.417 | 0.422 | **0.432** |

Table 11: 5-shot accuracy on the translated Global MMLU benchmark for extremely low-resource languages.

advantage across all three languages, with win rates of 39%–46% versus loss rates of 25%–31%. When compared to DoLa (Table 13), LGCD maintains a favorable win-loss margin in Zulu and Tswana, while showing comparable performance in Northern Sotho. Against model merging baselines TIES and SLERP (Tables 14 and 15), LGCD demonstrates a decisive advantage, particularly in Zulu where win rates exceed 85%.

| Language | Win | Tie | Lose |
|---|---|---|---|
| Zulu | 41% | 28% | 31% |
| Tswana | 46% | 29% | 25% |
| Northern Sotho | 39% | 31% | 30% |

Table 12: LGCD vs. LAM

| Language | Win | Tie | Lose |
|---|---|---|---|
| Zulu | 43% | 31% | 26% |
| Tswana | 40% | 33% | 27% |
| Northern Sotho | 37% | 28% | 35% |

Table 13: LGCD vs. DoLa

| Language | Win | Tie | Lose |
|---|---|---|---|
| Zulu | 85% | 12% | 3% |
| Tswana | 52% | 27% | 21% |
| Northern Sotho | 53% | 24% | 23% |

Table 14: LGCD vs. TIES

| Language | Win | Tie | Lose |
|---|---|---|---|
| Zulu | 86% | 12% | 2% |
| Tswana | 54% | 23% | 23% |
| Northern Sotho | 54% | 27% | 19% |

Table 15: LGCD vs. SLERP

## A.10 DIRECT COMPARISON OF LAM AND LGCD OUTPUTS

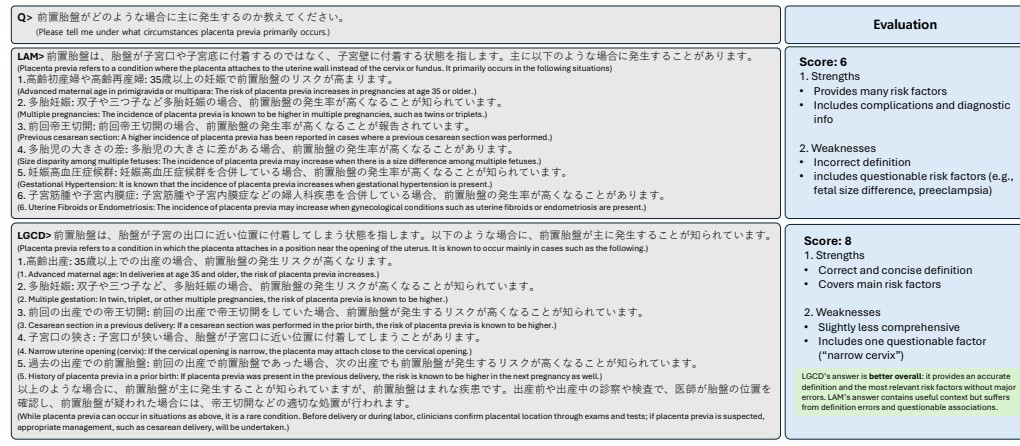

Figure 7: Direct comparison of LAM and LGCD outputs for a Japanese medical QA example (elyza/Llama-3-ELYZA-JP-8B) with evaluation

## A.11 ADAPTIVE CONFIDENCE THRESHOLD

We further study the robustness of LGCD to the choice of confidence threshold $\tau$ by introducing an *adaptive* thresholding mechanism that adjusts to the current decoding dynamics. At each decoding

| Lang. | Model | 0-shot | | 5-shot | |
|---|---|---|---|---|---|
| | | Adapt. | Heur. | Adapt. | Heur. |
| zh | hfl/llama-3-chinese-8b-instruct | 0.506 | 0.519 | 0.529 | 0.543 |
| zh | shenzhi-wang/Llama3-8B-Chinese-Chat | 0.504 | 0.502 | 0.544 | 0.543 |
| de | DiscoResearch/Llama3-DiscoLeo-Instruct-8B-v0.1 | 0.531 | 0.546 | 0.569 | 0.574 |
| pt | rhaymison/gemma-portuguese-luana-2b | 0.347 | 0.357 | 0.324 | 0.324 |
| ar | MohamedRashad/Arabic-Orpo-Llama-3-8B-Instruct | 0.464 | 0.465 | 0.479 | 0.481 |
| fa | PartAI/Dorna-Llama3-8B-Instruct | 0.424 | 0.423 | 0.466 | 0.466 |
| ja | tokyotech-llm/Llama-3-Swallow-8B-Instruct-v0.1 | 0.507 | 0.507 | 0.527 | 0.525 |
| ja | elyza/Llama-3-ELYZA-JP-8B | 0.507 | 0.509 | 0.528 | 0.528 |
| ko | KISTI-KONI/KONI-Llama3-8B-Instruct-20240729 | 0.483 | 0.490 | 0.505 | 0.511 |
| ko | MLP-KTLim/llama-3-Korean-Bllossom-8B | 0.477 | 0.479 | 0.498 | 0.501 |
| id | GoToCompany/llama-3-8b-cpt-sahabatai-v1-instruct | 0.537 | 0.527 | 0.574 | 0.566 |
| sw | Jacaranda/UlizaLlama3 | 0.388 | 0.399 | 0.391 | 0.409 |
| | **Average** | 0.473 | 0.477 | 0.495 | 0.498 |

Table 16: Comparison of adaptive and heuristic confidence thresholds $\tau$ for LGCD across languages and instruction-tuned LMs. Numbers denote task performance (higher is better).

step $t$, LGCD queries the LAM and obtains a confidence score

$$c_t = \max_y p_{\text{LAM}}(y \mid x, \hat{y}_{<t}),$$

i.e., the maximum softmax probability over candidate tokens. LGCD maintains a rolling history $H_t$ of recent confidence values with $|H_t| \leq 100$. Once at least 10 values have been accumulated ($|H_t| \geq 10$), we compute the adaptive threshold as

$$\tau_{\text{adaptive}} = \text{clip}\big(\mu_H - 0.5\sigma_H,\ 0.1,\ 0.9\big), \tag{10}$$

where $\mu_H$ and $\sigma_H$ denote the mean and standard deviation of the values in $H_t$, and $\text{clip}(x, a, b) = \min(\max(x, a), b)$ constrains the threshold to $[0.1, 0.9]$. Intuitively, $\tau$ decreases when the model becomes less certain (lower mean or higher variance in recent confidences), and increases when the model is consistently confident. The heuristic threshold used in our main experiments can be viewed as a resource-aware but *static* instantiation of this idea, whereas Eq. equation 10 adapts online to the LAM's confidence trajectory.

Table 16 compares the adaptive threshold $\tau_{\text{adaptive}}$ with our heuristic threshold across multiple languages and instruction-tuned LMs, under both 0-shot and 5-shot settings. The adaptive $\tau$ achieves performance that is slightly lower but nearly identical to the heuristic $\tau$ (typically within 0.002–0.01). Crucially, both thresholds consistently outperform strong baselines (PTM, LAM, DoLa, TIES, SLERP) across almost all languages (see Section 4), indicating that LGCD is robust to the precise choice of $\tau$ and that our simple heuristic threshold already provides an effective and language-resource-aware criterion.

## A.12 EVALUATION ON NON-KNOWLEDGE-INTENSIVE TASKS

While LGCD is designed primarily to mitigate factual degradation in language-adapted models, it is crucial to verify that it maintains the model's robustness across a broader range of general NLP tasks, encompassing creative, narrative, and fundamental reasoning capabilities. To this end, we conduct comprehensive evaluations on four diverse benchmarks encompassing creative, narrative, and fundamental reasoning capabilities: **XStoryCloze** (commonsense narrative reasoning), the **Aya Evaluation Suite** (open-ended conversational quality), **PAWS-X** (paraphrase identification), and **XNLI** (natural language inference).

**XStoryCloze (commonsense narrative reasoning) Lin et al. (2021b).** XStoryCloze is a multilingual extension of the StoryCloze Test, in which a model must select the most plausible ending to a four-sentence story. Because this task depends primarily on narrative coherence and commonsense reasoning rather than factual world knowledge, it serves as a direct test of whether LGCD harms reasoning or storytelling capabilities.

Each dataset supports only a specific subset of languages, so we evaluate in the languages where the benchmark's coverage overlaps with those of our adapted models. For XStoryCloze, this corresponds to Chinese, Arabic, Indonesian, and Swahili, and we follow the standard zero-shot evaluation setting used in prior work. We compare the pre-trained model (PTM), the language-adapted model (LAM), and LGCD.

| Lang. | Model | PTM | LAM | LGCD |
|---|---|---|---|---|
| zh | hfl/llama-3-chinese-8b-instruct | 0.689 | 0.682 | 0.695 |
| zh | shenzhi-wang/Llama3-8B-Chinese-Chat | 0.689 | 0.681 | 0.670 |
| ar | MohamedRashad/Arabic-Orpo-Llama-3-8B-Instruct | 0.609 | 0.608 | 0.587 |
| id | GoToCompany/llama3-8b-cpt-sahabatai-v1-instruct | 0.669 | 0.721 | 0.708 |
| sw | Jacaranda/UlizaLlama3 | 0.567 | 0.704 | 0.697 |
| | **Avg.** | 0.645 | 0.679 | 0.671 |

Table 17: Performance on XStoryCloze (commonsense narrative reasoning). Metrics are accuracy. LGCD matches or slightly trails LAM, and overall performance remains comparable, indicating that LGCD does not meaningfully harm reasoning or narrative capabilities.

Across models, LGCD closely tracks the LAM performance, with only small differences in accuracy (Table 17). Overall, LGCD preserves narrative understanding and commonsense reasoning while focusing on mitigating factual degradation.

**Aya Evaluation Suite (open-ended creative dialogue) Singh et al. (2024b).** The Aya Evaluation Suite contains 26,750 open-ended conversation-style prompts across seven languages, and is designed to evaluate creativity, coherence, and conversational quality rather than factual accuracy. As above, we evaluate only in languages that overlap with our adapted LAMs, namely Chinese, Arabic, and Portuguese. We perform GPT-4o-based pairwise evaluation comparing LGCD against LAM, and report the percentages of prompts where LGCD is preferred (Win), tied (Tie), or worse (Lose).

Across all tested languages, LGCD is preferred in roughly 55–58% of cases in open-ended dialogue generation (Table 18), indicating that LGCD does not harm diversity or expressiveness in generation and often yields more coherent and controlled outputs by avoiding over-confident hallucinated continuations.

| Lang. | Model | Win | Tie | Lose |
|---|---|---|---|---|
| zh | hfl/llama-3-chinese-8b-instruct | **58.0%** | 8.4% | 33.6% |
| zh | shenzhi-wang/Llama3-8B-Chinese-Chat | **56.4%** | 2.0% | 41.6% |
| ar | MohamedRashad/Arabic-Orpo-Llama-3-8B-Instruct | **54.4%** | 0.4% | 45.2% |
| pt | GoToCompany/llama3-8b-cpt-sahabatai-v1-instruct | **55.2%** | 3.2% | 41.6% |

Table 18: GPT-4o pairwise preference of LGCD vs. LAM on the Aya Evaluation Suite. "Win" indicates that LGCD is preferred over LAM for a given prompt, and "Lose" the opposite.

**PAWS-X (Paraphrase Identification).** PAWS-X is a cross-lingual adversarial dataset used to determine if two sentences are paraphrases. As shown in Table 19, LGCD achieves an average accuracy of 0.555 across Chinese, Japanese, and Korean models, slightly outperforming the LAM average of 0.548. Notably, LGCD shows consistent improvements in Korean models and one of the Chinese models, suggesting that our method's intervention is robust to paraphrase identification tasks.

| Lang. | Model | LAM | LGCD |
|---|---|---|---|
| zh | hfl/llama-3-chinese-8b-instruct | 0.528 | **0.553** |
| zh | shenzhi-wang/Llama3-8B-Chinese-Chat | **0.579** | 0.553 |
| ja | tokyotech-llm/Llama-3-Swallow-8B-Instruct-v0.1 | **0.569** | 0.559 |
| ja | elyza/Llama-3-ELYZA-JP-8B | 0.554 | **0.559** |
| ko | KISTI-KONI/KONI-Llama3-8B-Instruct-20240729 | 0.531 | **0.552** |
| ko | MLP-KTLim/llama-3-Korean-Bllossom-8B | 0.529 | **0.552** |
| | **Avg.** | 0.548 | **0.555** |

Table 19: Performance on PAWS-X. LGCD maintains or improves performance compared to LAM, demonstrating robustness in detecting paraphrases.

**XNLI (Natural Language Inference).** We also evaluated performance on XNLI, a task predicting textual entailment (entailment, contradiction, neutral). Table 20 shows that LGCD performs on par with LAM, with a slightly higher average score (0.331 vs. 0.329). In Arabic and Swahili, the performance remains identical, while in Chinese, results vary slightly depending on the base model but remain comparable.

| Lang. | Model | LAM | LGCD |
|-------|-------|-----|------|
| zh | hfl/llama-3-chinese-8b-instruct | **0.340** | 0.336 |
| zh | shenzhi-wang/Llama3-8B-Chinese-Chat | 0.336 | **0.347** |
| ar | MohamedRashad/Arabic-Orpo-Llama-3-8B-Instruct | **0.320** | **0.320** |
| sw | Jacaranda/UlizaLlama3 | **0.320** | **0.320** |
| | **Avg.** | 0.329 | **0.331** |

Table 20: Performance on XNLI (Natural Language Inference). LGCD preserves the reasoning capabilities of the adapted models, showing no significant degradation in NLI tasks.

## A.13 GENERAL-PURPOSE KOREAN MMLU EXPERIMENTS

We first examine a general-purpose SFT scenario in Korean to study how different fine-tuning strategies affect factual knowledge. We fine-tune `Llama-3-8B-Instruct` on an 11K Korean general-purpose SFT dataset used for training the KONI model, and evaluate zero-shot performance on the Korean portion of the Global MMLU benchmark.

Table 21 shows that all standard fine-tuning methods (full-parameter, attention-only, and FFN-only) suffer from catastrophic forgetting, performing worse than the original base model (0.437). Notably, attention-only fine-tuning yields the largest degradation (0.412), despite updating the parameters most directly associated with token-to-token interactions. In contrast, LGCD, which dynamically composes the language-adapted model (LAM) with the pre-trained model (PTM) at inference time, achieves the best performance (0.462), surpassing both the base model and all fine-tuning variants.

| Model | 0-shot (Global MMLU, ko) |
|-------|--------------------------|
| base model (no finetuning) | 0.437 |
| train (all layers) | 0.432 |
| train (attention only) | 0.412 |
| train (FFN only) | 0.431 |
| **LGCD (ours)** | **0.462** |

Table 21: Zero-shot performance on the Korean portion of Global MMLU after general-purpose SFT in Korean. All standard fine-tuning strategies underperform the base model, whereas LGCD, which composes the PTM and LAM at inference time, yields the highest accuracy.

These results indicate that freezing FFNs and updating only attention layers is not a reliable way to avoid catastrophic forgetting, and that dynamic inference-time composition via LGCD provides a more robust approach to preserving and enhancing factual knowledge. In the next subsection (Section A.14), we show that LGCD also improves accuracy in a domain-matched medical QA setting where fine-tuning itself is beneficial.

## A.14 DOMAIN-MATCHED MEDICAL QA EXPERIMENTS

We further evaluate LGCD in a task-aligned, domain-matched setting, where the adapted model is fine-tuned on the same downstream task on which it is evaluated. This setup tests whether LGCD remains beneficial even when the fine-tuned model already possesses strong domain knowledge.

**Experimental setup.** We consider three languages (Korean, Chinese, and Japanese) and evaluate on GlobalMedQA[2], a multilingual medical multiple-choice dataset designed to assess medical knowledge in LLMs. The base model is `meta-llama/Llama-3.1-8B-Instruct`. We compare the following adaptation variants: (i) full fine-tuning of all layers, (ii) fine-tuning attention layers only, (iii) fine-tuning feed-forward (FFN) layers only, and (iv) LGCD (ours), which composes the base and adapted models at inference time. All models are trained and evaluated on the same GlobalMedQA task for each language.

---

[2]`https://huggingface.co/datasets/mariocedo/GlobalMedQA`

| Model | Chinese | Japanese | Korean | Average |
|-------|---------|----------|--------|---------|
| base (no finetuning) | 55.21 | 41.70 | 48.02 | 48.31 |
| train (all layers) | 59.20 | 43.70 | 49.15 | 50.68 |
| train (attention only) | 57.40 | 42.90 | 47.59 | 49.30 |
| train (FFN only) | 59.40 | 43.00 | 49.09 | 50.50 |
| **LGCD (ours)** | **60.60** | **43.70** | **49.42** | **51.24** |

Table 22: Performance on GlobalMedQA in a domain-matched setting, where all adaptation variants are trained and evaluated on the same medical multiple-choice task. Metrics are accuracy (%). LGCD achieves the best average performance across the three languages.

Across Chinese, Japanese, and Korean, LGCD attains the highest average performance, outperforming full fine-tuning and layer-specific fine-tuning variants (Table 22). These results indicate that LGCD is effective not only when the adapted model forgets some truths, but also when the adapted model retains domain knowledge very well, and that composing the base and adapted models at inference time can provide additional gains even in fully domain-matched scenarios.

## A.15 TOP-K MASKING VS. ADAPTIVE PLAUSIBILITY CONSTRAINT

A natural alternative to LGCD's Top-K gating mechanism is the Adaptive Plausibility Constraint (APC) originally proposed in Contrastive Decoding (Li et al., 2023b), which retains only tokens whose probability exceeds a fraction $\alpha$ of the maximum probability:

$$\mathcal{V}_{\text{APC}}(x, \hat{y}_{<t}) = \left\{ y \in \mathcal{V} : p_{\text{PTM}}(y \mid x, \hat{y}_{<t}) \geq \alpha \cdot \max_{y'} p_{\text{PTM}}(y' \mid x, \hat{y}_{<t}) \right\}.$$

While APC adaptively narrows the candidate set when a single token dominates, Top-K masking provides a stable, language-agnostic candidate pool that preserves a fixed number of high-probability alternatives at every step. This stability is particularly beneficial in multilingual contrastive decoding, where the entropy landscape varies significantly across languages and language-specific models.

To validate this design choice, we replace Top-K masking in LGCD with APC-based masking (using $\alpha = 0.1$ following Li et al. (2023b)) and compare both variants on Global MMLU and long-form medical QA.

**Multiple-choice QA.** Table 23 reports Global MMLU accuracy across 12 language–model pairs under 0-shot and 5-shot settings. LGCD with Top-K masking consistently outperforms the APC variant, achieving $+2.8\%$ higher accuracy on average in the 0-shot setting and $+2.3\%$ in the 5-shot setting.

| Lang. | Model | 0-shot | | 5-shot | |
|-------|-------|--------|--------|--------|--------|
| | | APC | Top-K | APC | Top-K |
| zh | hfl/llama-3-chinese-8b-instruct | 0.460 | **0.519** | 0.500 | **0.543** |
| zh | shenzhi-wang/Llama3-8B-Chinese-Chat | 0.494 | **0.502** | 0.500 | **0.543** |
| de | DiscoResearch/Llama3-DiscoLeo-Instruct-8B-v0.1 | 0.471 | **0.546** | 0.534 | **0.574** |
| pt | rhaymison/gemma-portuguese-luana-2b | 0.286 | **0.357** | 0.307 | **0.324** |
| ar | MohamedRashad/Arabic-Orpo-Llama-3-8B-Instruct | 0.443 | **0.465** | 0.466 | **0.481** |
| fa | PartAI/Dorna-Llama3-8B-Instruct | **0.424** | 0.423 | 0.465 | **0.466** |
| ja | tokyotech-llm/Llama-3-Swallow-8B-Instruct-v0.1 | 0.494 | **0.507** | 0.522 | **0.525** |
| ja | elyza/Llama-3-ELYZA-JP-8B | 0.501 | **0.509** | 0.518 | **0.528** |
| ko | KISTI-KONI/KONI-Llama3-8B-Instruct-20240729 | 0.458 | **0.490** | 0.481 | **0.511** |
| ko | MLP-KTLim/llama-3-Korean-Bllossom-8B | 0.464 | **0.479** | 0.487 | **0.501** |
| id | GoToCompany/llama3-8b-cpt-sahabatai-v1-instruct | **0.535** | 0.527 | **0.570** | 0.566 |
| sw | Jacaranda/UlizaLlama3 | 0.352 | **0.399** | 0.353 | **0.409** |
| | **Average** | 0.449 | **0.477** | 0.475 | **0.498** |

Table 23: Global MMLU accuracy comparing LGCD with Top-K masking against LGCD with APC-based masking across languages and instruction-tuned LMs. Bold indicates the better variant for each row.

**Long-form generation.** Table 24 presents pairwise GPT-4o evaluation results on long-form medical QA, comparing outputs generated by LGCD (Top-K) against those from LGCD (APC). On average, LGCD with Top-K masking wins $53.3\%$ of comparisons versus $44.7\%$ for the APC variant,

with only 2.0% ties. The Top-K variant is preferred in 8 out of 12 language–model pairs, confirming that the fixed-size candidate pool leads to more fluent and informative multilingual generations. These results suggest that, in the multilingual contrastive decoding setting, the consistent candidate coverage provided by Top-K masking is more effective than APC's adaptive but potentially over-aggressive pruning strategy.

| Lang. | Model | Win | Tie | Lose |
|---|---|---|---|---|
| zh | hfl/llama-3-chinese-8b-instruct | **61.0** | 0.7 | 38.3 |
| zh | shenzhi-wang/Llama3-8B-Chinese-Chat | 43.0 | 5.7 | **51.3** |
| de | DiscoResearch/Llama3-DiscoLeo-Instruct-8B-v0.1 | **58.9** | 0.0 | 41.2 |
| pt | rhaymison/gemma-portuguese-luana-2b | **61.0** | 1.0 | 38.0 |
| ar | MohamedRashad/Arabic-Orpo-Llama-3-8B-Instruct | 47.7 | 1.0 | **51.3** |
| fa | PartAI/Dorna-Llama3-8B-Instruct | **51.9** | 1.9 | 46.3 |
| ja | tokyotech-llm/Llama-3-Swallow-8B-Instruct-v0.1 | **57.5** | 1.0 | 41.5 |
| ja | elyza/Llama-3-ELYZA-JP-8B | 47.0 | 0.0 | **53.0** |
| ko | KISTI-KONI/KONI-Llama3-8B-Instruct-20240729 | **53.6** | 0.4 | 46.0 |
| ko | MLP-KTLim/llama-3-Korean-Bllossom-8B | 46.0 | 3.4 | **50.6** |
| id | GoToCompany/llama3-8b-cpt-sahabatai-v1-instruct | **57.3** | 5.4 | 37.3 |
| sw | Jacaranda/UlizaLlama3 | **55.4** | 3.1 | 41.5 |
| | Average | **53.3** | 2.0 | 44.7 |

Table 24: Pairwise GPT-4o evaluation on long-form medical QA: LGCD (Top-K) vs. LGCD (APC). Win/Tie/Lose (%) are reported from the perspective of the Top-K variant. Bold indicates the higher value between Win and Lose for each row.

## A.16 ADDITIONAL MODEL-MERGING RESULTS FOR GLOBAL MMLU

In Table 2 we report two representative merge methods, TIES and SLERP. This appendix extends the analysis to fourteen widely used model-merging methods, including arcee_fusion Goddard et al. (2024), breadcrumbs Davari & Belilovsky (2024), breadcrumbs_ties Davari & Belilovsky (2024), dare_linear Yu et al. (2024), dare_ties Yu et al. (2024), della Deep et al. (2024), della_linear Deep et al. (2024), karcher, linear Wortsman et al. (2022), multislerp, sce Wan et al. (2025), slerp Shoemake (1985), task_arithmetic Ilharco et al. (2022), and ties Yadav et al. (2023b). All evaluations use the same setup as described in the main text. For method definitions and references, please refer to the official MergeKit documentation: `https://github.com/arcee-ai/mergekit/blob/main/docs/merge_methods.md`.

| Lang. | Model | Method | 0-shot | 5-shot |
|---|---|---|---|---|
| zh | hfl/llama-3-chinese-8b-instruct | arcee˙fusion | 0.471 | 0.510 |
| zh | hfl/llama-3-chinese-8b-instruct | breadcrumbs | 0.490 | 0.532 |
| zh | hfl/llama-3-chinese-8b-instruct | breadcrumbs˙ties | 0.490 | 0.532 |
| zh | hfl/llama-3-chinese-8b-instruct | dare˙linear | 0.460 | 0.475 |
| zh | hfl/llama-3-chinese-8b-instruct | dare˙ties | 0.469 | 0.497 |
| zh | hfl/llama-3-chinese-8b-instruct | della | 0.312 | 0.309 |
| zh | hfl/llama-3-chinese-8b-instruct | della˙linear | 0.346 | 0.368 |
| zh | hfl/llama-3-chinese-8b-instruct | karcher | 0.488 | 0.529 |
| zh | hfl/llama-3-chinese-8b-instruct | linear | 0.490 | 0.531 |
| zh | hfl/llama-3-chinese-8b-instruct | multislerp | 0.441 | 0.496 |
| zh | hfl/llama-3-chinese-8b-instruct | sce | 0.441 | 0.498 |
| zh | hfl/llama-3-chinese-8b-instruct | slerp | 0.492 | 0.531 |
| zh | hfl/llama-3-chinese-8b-instruct | task˙arithmetic | 0.490 | 0.532 |
| zh | hfl/llama-3-chinese-8b-instruct | ties | 0.475 | 0.524 |
| zh | shenzhi-wang/Llama3-8B-Chinese-Chat | arcee˙fusion | 0.480 | 0.527 |
| zh | shenzhi-wang/Llama3-8B-Chinese-Chat | breadcrumbs | 0.482 | 0.530 |
| zh | shenzhi-wang/Llama3-8B-Chinese-Chat | breadcrumbs˙ties | 0.482 | 0.530 |
| zh | shenzhi-wang/Llama3-8B-Chinese-Chat | dare˙linear | 0.483 | 0.529 |
| zh | shenzhi-wang/Llama3-8B-Chinese-Chat | dare˙ties | 0.483 | 0.529 |
| zh | shenzhi-wang/Llama3-8B-Chinese-Chat | della | 0.473 | 0.525 |
| zh | shenzhi-wang/Llama3-8B-Chinese-Chat | della˙linear | 0.472 | 0.524 |
| zh | shenzhi-wang/Llama3-8B-Chinese-Chat | karcher | 0.482 | 0.529 |
| zh | shenzhi-wang/Llama3-8B-Chinese-Chat | linear | 0.482 | 0.529 |
| zh | shenzhi-wang/Llama3-8B-Chinese-Chat | multislerp | 0.476 | 0.526 |
| zh | shenzhi-wang/Llama3-8B-Chinese-Chat | sce | 0.475 | 0.526 |

Table 25 – continued from previous page

| Lang. | Model | Method | 0-shot | 5-shot |
|---|---|---|---|---|
| zh | shenzhi-wang/Llama3-8B-Chinese-Chat | slerp | 0.482 | 0.529 |
| zh | shenzhi-wang/Llama3-8B-Chinese-Chat | task˙arithmetic | 0.482 | 0.530 |
| zh | shenzhi-wang/Llama3-8B-Chinese-Chat | ties | 0.505 | 0.550 |
| de | DiscoResearch/Llama3-DiscoLeo-Instruct-8B-v0.1 | arcee˙fusion | 0.505 | 0.556 |
| de | DiscoResearch/Llama3-DiscoLeo-Instruct-8B-v0.1 | breadcrumbs | 0.515 | 0.567 |
| de | DiscoResearch/Llama3-DiscoLeo-Instruct-8B-v0.1 | breadcrumbs˙ties | 0.515 | 0.567 |
| de | DiscoResearch/Llama3-DiscoLeo-Instruct-8B-v0.1 | dare˙linear | 0.505 | 0.556 |
| de | DiscoResearch/Llama3-DiscoLeo-Instruct-8B-v0.1 | dare˙ties | 0.504 | 0.558 |
| de | DiscoResearch/Llama3-DiscoLeo-Instruct-8B-v0.1 | della | 0.404 | 0.503 |
| de | DiscoResearch/Llama3-DiscoLeo-Instruct-8B-v0.1 | della˙linear | 0.415 | 0.512 |
| de | DiscoResearch/Llama3-DiscoLeo-Instruct-8B-v0.1 | karcher | 0.514 | 0.563 |
| de | DiscoResearch/Llama3-DiscoLeo-Instruct-8B-v0.1 | linear | 0.516 | 0.567 |
| de | DiscoResearch/Llama3-DiscoLeo-Instruct-8B-v0.1 | multislerp | 0.463 | 0.532 |
| de | DiscoResearch/Llama3-DiscoLeo-Instruct-8B-v0.1 | sce | 0.463 | 0.532 |
| de | DiscoResearch/Llama3-DiscoLeo-Instruct-8B-v0.1 | slerp | 0.516 | 0.565 |
| de | DiscoResearch/Llama3-DiscoLeo-Instruct-8B-v0.1 | task˙arithmetic | 0.515 | 0.567 |
| de | DiscoResearch/Llama3-DiscoLeo-Instruct-8B-v0.1 | ties | 0.502 | 0.565 |
| pt | rhaymison/gemma-portuguese-luana-2b | arcee˙fusion | 0.324 | 0.301 |
| pt | rhaymison/gemma-portuguese-luana-2b | breadcrumbs | 0.332 | 0.322 |
| pt | rhaymison/gemma-portuguese-luana-2b | breadcrumbs˙ties | 0.332 | 0.322 |
| pt | rhaymison/gemma-portuguese-luana-2b | dare˙linear | 0.327 | 0.314 |
| pt | rhaymison/gemma-portuguese-luana-2b | dare˙ties | 0.323 | 0.318 |
| pt | rhaymison/gemma-portuguese-luana-2b | della | 0.273 | 0.277 |
| pt | rhaymison/gemma-portuguese-luana-2b | della˙linear | 0.258 | 0.273 |
| pt | rhaymison/gemma-portuguese-luana-2b | karcher | 0.335 | 0.323 |
| pt | rhaymison/gemma-portuguese-luana-2b | linear | 0.332 | 0.322 |
| pt | rhaymison/gemma-portuguese-luana-2b | multislerp | 0.294 | 0.301 |
| pt | rhaymison/gemma-portuguese-luana-2b | sce | 0.293 | 0.301 |
| pt | rhaymison/gemma-portuguese-luana-2b | slerp | 0.334 | 0.323 |
| pt | rhaymison/gemma-portuguese-luana-2b | task˙arithmetic | 0.332 | 0.322 |
| pt | rhaymison/gemma-portuguese-luana-2b | ties | 0.334 | 0.327 |
| ar | MohamedRashad/Arabic-Orpo-Llama-3-8B-Instruct | arcee˙fusion | 0.413 | 0.466 |
| ar | MohamedRashad/Arabic-Orpo-Llama-3-8B-Instruct | breadcrumbs | 0.418 | 0.466 |
| ar | MohamedRashad/Arabic-Orpo-Llama-3-8B-Instruct | breadcrumbs˙ties | 0.418 | 0.466 |
| ar | MohamedRashad/Arabic-Orpo-Llama-3-8B-Instruct | dare˙linear | 0.420 | 0.466 |
| ar | MohamedRashad/Arabic-Orpo-Llama-3-8B-Instruct | dare˙ties | 0.418 | 0.466 |
| ar | MohamedRashad/Arabic-Orpo-Llama-3-8B-Instruct | della | 0.407 | 0.453 |
| ar | MohamedRashad/Arabic-Orpo-Llama-3-8B-Instruct | della˙linear | 0.406 | 0.453 |
| ar | MohamedRashad/Arabic-Orpo-Llama-3-8B-Instruct | karcher | 0.417 | 0.465 |
| ar | MohamedRashad/Arabic-Orpo-Llama-3-8B-Instruct | linear | 0.418 | 0.466 |
| ar | MohamedRashad/Arabic-Orpo-Llama-3-8B-Instruct | multislerp | 0.408 | 0.454 |
| ar | MohamedRashad/Arabic-Orpo-Llama-3-8B-Instruct | sce | 0.406 | 0.454 |
| ar | MohamedRashad/Arabic-Orpo-Llama-3-8B-Instruct | slerp | 0.418 | 0.466 |
| ar | MohamedRashad/Arabic-Orpo-Llama-3-8B-Instruct | task˙arithmetic | 0.418 | 0.466 |
| ar | MohamedRashad/Arabic-Orpo-Llama-3-8B-Instruct | ties | 0.433 | 0.476 |
| fa | PartAI/Dorna-Llama3-8B-Instruct | arcee˙fusion | 0.400 | 0.449 |
| fa | PartAI/Dorna-Llama3-8B-Instruct | breadcrumbs | 0.401 | 0.449 |
| fa | PartAI/Dorna-Llama3-8B-Instruct | breadcrumbs˙ties | 0.401 | 0.449 |
| fa | PartAI/Dorna-Llama3-8B-Instruct | dare˙linear | 0.400 | 0.448 |
| fa | PartAI/Dorna-Llama3-8B-Instruct | dare˙ties | 0.399 | 0.449 |
| fa | PartAI/Dorna-Llama3-8B-Instruct | della | 0.400 | 0.450 |
| fa | PartAI/Dorna-Llama3-8B-Instruct | della˙linear | 0.399 | 0.449 |
| fa | PartAI/Dorna-Llama3-8B-Instruct | karcher | 0.398 | 0.448 |
| fa | PartAI/Dorna-Llama3-8B-Instruct | linear | 0.400 | 0.449 |
| fa | PartAI/Dorna-Llama3-8B-Instruct | multislerp | 0.400 | 0.450 |
| fa | PartAI/Dorna-Llama3-8B-Instruct | sce | 0.402 | 0.449 |
| fa | PartAI/Dorna-Llama3-8B-Instruct | slerp | 0.400 | 0.449 |
| fa | PartAI/Dorna-Llama3-8B-Instruct | task˙arithmetic | 0.401 | 0.449 |
| fa | PartAI/Dorna-Llama3-8B-Instruct | ties | 0.429 | 0.471 |
| ja | tokyotech-llm/Llama-3-Swallow-8B-Instruct-v0.1 | arcee˙fusion | 0.458 | 0.502 |
| ja | tokyotech-llm/Llama-3-Swallow-8B-Instruct-v0.1 | breadcrumbs | 0.473 | 0.515 |

Table 25 – continued from previous page

| Lang. | Model | Method | 0-shot | 5-shot |
|-------|-------|--------|--------|--------|
| ja | tokyotech-llm/Llama-3-Swallow-8B-Instruct-v0.1 | breadcrumbs˙ties | 0.473 | 0.515 |
| ja | tokyotech-llm/Llama-3-Swallow-8B-Instruct-v0.1 | dare˙linear | 0.456 | 0.495 |
| ja | tokyotech-llm/Llama-3-Swallow-8B-Instruct-v0.1 | dare˙ties | 0.461 | 0.508 |
| ja | tokyotech-llm/Llama-3-Swallow-8B-Instruct-v0.1 | della | 0.429 | 0.478 |
| ja | tokyotech-llm/Llama-3-Swallow-8B-Instruct-v0.1 | della˙linear | 0.427 | 0.481 |
| ja | tokyotech-llm/Llama-3-Swallow-8B-Instruct-v0.1 | karcher | 0.472 | 0.519 |
| ja | tokyotech-llm/Llama-3-Swallow-8B-Instruct-v0.1 | linear | 0.473 | 0.516 |
| ja | tokyotech-llm/Llama-3-Swallow-8B-Instruct-v0.1 | multislerp | 0.456 | 0.510 |
| ja | tokyotech-llm/Llama-3-Swallow-8B-Instruct-v0.1 | sce | 0.455 | 0.510 |
| ja | tokyotech-llm/Llama-3-Swallow-8B-Instruct-v0.1 | slerp | 0.474 | 0.517 |
| ja | tokyotech-llm/Llama-3-Swallow-8B-Instruct-v0.1 | task˙arithmetic | 0.473 | 0.515 |
| ja | tokyotech-llm/Llama-3-Swallow-8B-Instruct-v0.1 | ties | 0.482 | 0.528 |
| ja | elyza/Llama-3-ELYZA-JP-8B | arcee˙fusion | 0.440 | 0.472 |
| ja | elyza/Llama-3-ELYZA-JP-8B | breadcrumbs | 0.471 | 0.505 |
| ja | elyza/Llama-3-ELYZA-JP-8B | breadcrumbs˙ties | 0.471 | 0.505 |
| ja | elyza/Llama-3-ELYZA-JP-8B | dare˙linear | 0.449 | 0.470 |
| ja | elyza/Llama-3-ELYZA-JP-8B | dare˙ties | 0.453 | 0.482 |
| ja | elyza/Llama-3-ELYZA-JP-8B | della | 0.411 | 0.431 |
| ja | elyza/Llama-3-ELYZA-JP-8B | della˙linear | 0.387 | 0.424 |
| ja | elyza/Llama-3-ELYZA-JP-8B | karcher | 0.474 | 0.513 |
| ja | elyza/Llama-3-ELYZA-JP-8B | linear | 0.469 | 0.504 |
| ja | elyza/Llama-3-ELYZA-JP-8B | multislerp | 0.452 | 0.488 |
| ja | elyza/Llama-3-ELYZA-JP-8B | sce | 0.451 | 0.488 |
| ja | elyza/Llama-3-ELYZA-JP-8B | slerp | 0.470 | 0.505 |
| ja | elyza/Llama-3-ELYZA-JP-8B | task˙arithmetic | 0.471 | 0.505 |
| ja | elyza/Llama-3-ELYZA-JP-8B | ties | 0.470 | 0.506 |
| ko | KISTI-KONI/KONI-Llama3-8B-Instruct-20240729 | arcee˙fusion | 0.439 | 0.486 |
| ko | KISTI-KONI/KONI-Llama3-8B-Instruct-20240729 | breadcrumbs | 0.452 | 0.503 |
| ko | KISTI-KONI/KONI-Llama3-8B-Instruct-20240729 | breadcrumbs˙ties | 0.452 | 0.503 |
| ko | KISTI-KONI/KONI-Llama3-8B-Instruct-20240729 | dare˙linear | 0.438 | 0.482 |
| ko | KISTI-KONI/KONI-Llama3-8B-Instruct-20240729 | dare˙ties | 0.442 | 0.485 |
| ko | KISTI-KONI/KONI-Llama3-8B-Instruct-20240729 | della | 0.409 | 0.443 |
| ko | KISTI-KONI/KONI-Llama3-8B-Instruct-20240729 | della˙linear | 0.396 | 0.444 |
| ko | KISTI-KONI/KONI-Llama3-8B-Instruct-20240729 | karcher | 0.450 | 0.500 |
| ko | KISTI-KONI/KONI-Llama3-8B-Instruct-20240729 | linear | 0.452 | 0.502 |
| ko | KISTI-KONI/KONI-Llama3-8B-Instruct-20240729 | multislerp | 0.436 | 0.476 |
| ko | KISTI-KONI/KONI-Llama3-8B-Instruct-20240729 | sce | 0.436 | 0.477 |
| ko | KISTI-KONI/KONI-Llama3-8B-Instruct-20240729 | slerp | 0.451 | 0.503 |
| ko | KISTI-KONI/KONI-Llama3-8B-Instruct-20240729 | task˙arithmetic | 0.452 | 0.503 |
| ko | KISTI-KONI/KONI-Llama3-8B-Instruct-20240729 | ties | 0.469 | 0.507 |
| ko | MLP-KTLim/llama-3-Korean-Bllossom-8B | arcee˙fusion | 0.360 | 0.431 |
| ko | MLP-KTLim/llama-3-Korean-Bllossom-8B | breadcrumbs | 0.407 | 0.470 |
| ko | MLP-KTLim/llama-3-Korean-Bllossom-8B | breadcrumbs˙ties | 0.407 | 0.470 |
| ko | MLP-KTLim/llama-3-Korean-Bllossom-8B | dare˙linear | 0.374 | 0.449 |
| ko | MLP-KTLim/llama-3-Korean-Bllossom-8B | dare˙ties | 0.373 | 0.450 |
| ko | MLP-KTLim/llama-3-Korean-Bllossom-8B | della | 0.310 | 0.392 |
| ko | MLP-KTLim/llama-3-Korean-Bllossom-8B | della˙linear | 0.307 | 0.369 |
| ko | MLP-KTLim/llama-3-Korean-Bllossom-8B | karcher | 0.215 | 0.242 |
| ko | MLP-KTLim/llama-3-Korean-Bllossom-8B | linear | 0.408 | 0.469 |
| ko | MLP-KTLim/llama-3-Korean-Bllossom-8B | multislerp | 0.352 | 0.433 |
| ko | MLP-KTLim/llama-3-Korean-Bllossom-8B | sce | 0.353 | 0.432 |
| ko | MLP-KTLim/llama-3-Korean-Bllossom-8B | slerp | 0.392 | 0.466 |
| ko | MLP-KTLim/llama-3-Korean-Bllossom-8B | task˙arithmetic | 0.407 | 0.470 |
| ko | MLP-KTLim/llama-3-Korean-Bllossom-8B | ties | 0.380 | 0.469 |
| id | GoToCompany/llama3-8b-cpt-sahabatai-v1-instruct | arcee˙fusion | 0.466 | 0.500 |
| id | GoToCompany/llama3-8b-cpt-sahabatai-v1-instruct | breadcrumbs | 0.511 | 0.562 |
| id | GoToCompany/llama3-8b-cpt-sahabatai-v1-instruct | breadcrumbs˙ties | 0.511 | 0.562 |
| id | GoToCompany/llama3-8b-cpt-sahabatai-v1-instruct | dare˙linear | 0.490 | 0.539 |
| id | GoToCompany/llama3-8b-cpt-sahabatai-v1-instruct | dare˙ties | 0.486 | 0.517 |
| id | GoToCompany/llama3-8b-cpt-sahabatai-v1-instruct | della | 0.456 | 0.509 |
| id | GoToCompany/llama3-8b-cpt-sahabatai-v1-instruct | della˙linear | 0.463 | 0.461 |

Continued on next page

Table 25 – continued from previous page

| Lang. | Model | Method | 0-shot | 5-shot |
|---|---|---|---|---|
| id | GoToCompany/llama3-8b-cpt-sahabatai-v1-instruct | karcher | 0.511 | 0.558 |
| id | GoToCompany/llama3-8b-cpt-sahabatai-v1-instruct | linear | 0.511 | 0.560 |
| id | GoToCompany/llama3-8b-cpt-sahabatai-v1-instruct | multislerp | 0.505 | 0.560 |
| id | GoToCompany/llama3-8b-cpt-sahabatai-v1-instruct | sce | 0.507 | 0.559 |
| id | GoToCompany/llama3-8b-cpt-sahabatai-v1-instruct | slerp | 0.512 | 0.562 |
| id | GoToCompany/llama3-8b-cpt-sahabatai-v1-instruct | task˙arithmetic | 0.511 | 0.562 |
| id | GoToCompany/llama3-8b-cpt-sahabatai-v1-instruct | ties | 0.528 | 0.571 |
| sw | Jacaranda/UlizaLlama3 | arcee˙fusion | 0.321 | 0.427 |
| sw | Jacaranda/UlizaLlama3 | breadcrumbs | 0.395 | 0.427 |
| sw | Jacaranda/UlizaLlama3 | breadcrumbs˙ties | 0.395 | 0.275 |
| sw | Jacaranda/UlizaLlama3 | dare˙linear | 0.282 | 0.336 |
| sw | Jacaranda/UlizaLlama3 | dare˙ties | 0.298 | 0.255 |
| sw | Jacaranda/UlizaLlama3 | della | 0.260 | 0.227 |
| sw | Jacaranda/UlizaLlama3 | della˙linear | 0.245 | 0.422 |
| sw | Jacaranda/UlizaLlama3 | karcher | 0.392 | 0.426 |
| sw | Jacaranda/UlizaLlama3 | linear | 0.396 | 0.350 |
| sw | Jacaranda/UlizaLlama3 | multislerp | 0.340 | 0.352 |
| sw | Jacaranda/UlizaLlama3 | sce | 0.340 | 0.426 |
| sw | Jacaranda/UlizaLlama3 | slerp | 0.397 | 0.427 |
| sw | Jacaranda/UlizaLlama3 | task˙arithmetic | 0.395 | 0.381 |
| sw | Jacaranda/UlizaLlama3 | ties | 0.380 | 0.381 |

Table 26: Results of fourteen model-merging methods on Global MMLU under 0-shot and 5-shot settings.

