# OpenReview forum: "Leveraging Pretrained Knowledge at Inference Time: LoRA-Gated Contrastive Decoding for Multilingual Factual Language Generation in Adapted LLMs"
_ICLR.cc/2026/Conference — ICLR 2026 Poster_

### Official Review · Reviewer_GJXc · 2025-10-30

**Soundness:** 3
**Presentation:** 3
**Contribution:** 3
**Rating:** 6
**Confidence:** 3

**Summary:**

This paper proposes LoRA-Gated Contrastive Decoding (LGCD), an inference-time framework designed to improve factual accuracy in language-adapted LLMs that have undergone continual pretraining or instruction tuning. The method approximates the pretrained model’s factual knowledge by extracting low-rank updates from FFN layers, and dynamically switches between the adapted model and the approximated pretrained model based on token-level confidence. Experiments across multiple languages show improvements in factuality without additional training or access to pretraining data.

**Strengths:**

- The method does not require access to the original pretraining data, nor retraining or additional model parameters.

- Only extracting weights from the FFN layers makes implementing the method lightweight.

- The confidence-based gating is intuitive and allows the method to selectively intervene only when needed, balancing fluency and factuality.

- Experiments across multiple languages demonstrate general applicability.

**Weaknesses:**

- Incomplete evaluation: The paper motivates the problem by arguing that continual pretraining and naive LoRA fine-tuning lead to catastrophic forgetting; however, their evaluation does not include LoRA fine-tuned models as a baseline, even though this is a directly relevant and stronger baseline than the model-merging baselines used. This omission weakens the empirical evidence for the claimed benefit of the proposed method.

- The evaluation is limited to QA-style benchmarks, which constrains the generalizability of the findings. Additional tasks would help demonstrate broader applicability, while the Multi-FAct score eval should be expanded.

- The work focuses primarily on mid- and high-resource languages. Nigatu et al. (2024) classify Swahili as a low-resource language; Swahili has recently become closer to a mid-resource language due to expanding corpora and tooling. Similarly, the other languages evaluated are not particularly low-resource, which limits the claim that the method addresses low-resource multilingual adaptation. See Nigatu et al. (2024) for some low-resource languages.

$~$

### **References**

Nigatu, Hellina Hailu, et al. *"The Zeno's Paradox of Low-Resource 'Languages'."* arXiv preprint arXiv:2410.20817 (2024).

**Questions:**

1. How do you think your method will compare to replacing equations 3 and 4 with actual LoRA CPT on the FFN layers instead of SVD-derived $A_\ell\$ and $B_\ell\$​? Can you show that?

2. How exactly did you empirically show that your method mitigates hallucination?

3. Can you provide results for benchmarks beyond QA tasks? The Multi-Fact eval (Table 4) only shows results for your method and PTM, which is limiting, as Figure 2 shows that other methods are also competitive.

4. With LGCD, do the pretrained and adapted models have to use the same architecture? Table 1 indicates that you used models with a mix of Gemma and Llama architecture.

5. At inference time, do you deploy both PTM and LAM while alternating the decoding with LGCD? A short description of that part will help.

6. Do you have an idea of what happened to Japanese in Figure 2?

---

> ### Author Response · Authors · 2025-11-19
> **Author Rebuttal - W1**
>
> Thank you for the comment. The suggestion to include LoRA fine-tuned models as a direct baseline is an excellent one, and we agree that this comparison is essential for a complete evaluation.
>
> Our paper's initial framing was heavily motivated by a common, practical "real-world" scenario: a user or researcher downloads a publicly-available, language-adapted model (LAM) from a repository like Hugging Face. In the vast majority of these cases, the original continual pretraining or instruction tuning data is not disclosed, making it impossible to train a new, directly comparable LoRA baseline. Our training-free LGCD framework was designed as a flexible and realistic solution precisely for this common setting, allowing for the enhancement of any existing LAM without needing its original data or retraining.
>
> However, we fully agree with the reviewer that for a robust empirical validation, a direct comparison against a LoRA-trained baseline is crucial. Following your valuable suggestion, we have conducted two new sets of experiments to simulate both general-purpose and domain-specific adaptation scenarios using LoRA. These results demonstrate LGCD's effectiveness in both mitigating catastrophic forgetting and enhancing already-strong models.
>
> ### **(1) New Experiment 1: General-Purpose SFT (Korean)**
>
> We first evaluated a general-purpose SFT scenario. We fine-tuned the  llama3-8B-instruct model on a 11K Korean general-purpose SFT dataset that used for training KONI model and evaluated it on the Global MMLU (ko) benchmark (0-shot).
>
> As shown in the below table, all fine-tuning methods (all-layers, ffn-only, and attention-only) suffered from catastrophic forgetting, performing worse than the original base model (0.437). Notably, the train (attention-only) method, as suggested by the reviewer, showed the most significant performance degradation (0.412). In contrast, our LGCD method, by dynamically leveraging the LAM and the PTM, achieved the highest performance (0.462).
>
> | **Model** | **0-shot (Global MMLU - ko)** |
> | --- | --- |
> | base model (w/o finetuning) | 0.437 |
> | train (all layers) | 0.432 |
> | train (attention only) | 0.412 |
> | train (ffn only) | 0.431 |
> | **LGCD (Ours)** | **0.462** |
>
> ### **(2) New Experiment 2: Domain-Matched SFT (Multilingual Medical QA)**
>
> To demonstrate that LGCD is not just a "remedy" for forgetting but also an "enhancement" for well-adapted models, we ran a second experiment in a "domain-matched" setting. We fine-tuned the meta-llama/Llama-3.1-8B-Instruct model on the GlobalMedQA dataset (a multilingual medical MCQA benchmark) and evaluated it on the same task.
>
> In this scenario, the LoRA fine-tuned models (LAMs) expectedly outperformed the base model (avg. 48.31). However, our LGCD method (applied on top of the train (all layers) LAM) still achieved the best average performance (51.24). This result is crucial, as it shows that LGCD provides significant factuality gains even when the LAM is well-adapted and possesses strong, newly-acquired domain knowledge.
>
> | **Model** | **Chinese** | **Japanese** | **Korean** | **Average** |
> | --- | --- | --- | --- | --- |
> | base (no finetuning) | 55.21 | 41.70 | 48.02 | 48.31 |
> | LoRA fine-tuned (all layers) | 59.20 | 43.70 | 49.15 | 50.68 |
> | LoRA fine-tuned (attention only) | 57.40 | 42.90 | 47.59 | 49.30 |
> | LoRA fine-tuned (FFN only) | 59.40 | 43.00 | 49.09 | 50.50 |
> | **LGCD (Ours)** | **60.60** | **43.70** | **49.42** | **51.24** |
>
> These new experiments, which we have added to the Appendix, confirm that a LoRA fine-tuned model is indeed a relevant and strong baseline. They also empirically validate our method's key advantages: LGCD effectively mitigates catastrophic forgetting where it occurs (Exp 1) and further enhances factuality even when fine-tuning is beneficial (Exp 2).

---

> ### Author Response · Authors · 2025-11-19
> **Author Rebuttal - W2**
>
> Thank you for the comment. We agree with your comment regarding the need to demonstrate broader generalizability. while our paper's primary motivation is mitigating the factual degradation evident in knowledge-intensive QA tasks, it is essential to demonstrate LGCD's broader applicability and ensure it does not harm other core language capabilities, such as reasoning or creativity.
>
> To address your concern, we have conducted new evaluations on two
> widely-used non-factual benchmarks—XStoryCloze (for commonsense narrative reasoning) and the Aya Evaluation Suite (for open-ended creative dialogue). These benchmarks specifically test the model's capabilities in areas distinct from knowledge-based QA.
>
> These results confirm that LGCD, while optimized for factuality, preserves, and in several cases improves, performance on these non-factual tasks.
>
> ### **(1) Evaluation on XStoryCloze (Commonsense Narrative Reasoning)**
>
> We first evaluated LGCD on XStoryCloze, a benchmark that tests narrative understanding by requiring the model to select the most plausible ending for a four-sentence story. This task relies on commonsense reasoning, not the factual world knowledge targeted by LGCD. We evaluated on the languages in our set that overlap with the benchmark (Chinese, Arabic, Indonesian, and Swahili) in a zero-shot setting.
>
> | **Lang.** | **Model** | **PTM** | **LAM** | **LGCD** |
> | --- | --- | --- | --- | --- |
> | zh | hfl/llama-3-chinese-8b-instruct | 0.689 | 0.682 | 0.695 |
> | zh | shenzhi-wang/Llama3-8B-Chinese-Chat | 0.689 | 0.681 | 0.670 |
> | ar | MohamedRashad/Arabic-Orpo-Llama-3-8B-Instruct | 0.609 | 0.608 | 0.587 |
> | id | GoToCompany/llama3-8b-cpt-sahabatai-v1-instruct | 0.669 | 0.721 | 0.708 |
> | sw | Jacaranda/UlizaLlama3 | 0.567 | 0.704 | 0.697 |
> | Avg. |  | 0.645 | 0.679 | 0.671 |
>
> The results show that LGCD's performance (Avg. 0.671) is highly comparable to the baseline LAM (Avg. 0.679). It demonstrates that our method, which is explicitly designed to enhance factuality, does not harm and largely maintains the model's core performance on non-factual, commonsense reasoning tasks.
>
> ### **(2) Aya Evaluation Suite (Open-ended Creative Dialogue)**
>
> Aya benchmark contains 26,750 open-ended conversation-style prompts across 7 languages. This benchmark evaluates creativity, coherence, and conversational quality, not factual accuracy. As with XStoryCloze, we evaluate only the languages that overlap with our adapted LAMs (Chinese, Arabic, Portuguese). We performed GPT-4o pairwise evaluation comparing LGCD vs. LAM:
>
> | Lang. | Win | Tie | Lose |
> | --- | --- | --- | --- |
> | zh | **58.0%** | 8.4% | 33.6% |
> | zh | **56.4%** | 2.0% | 41.6% |
> | ar | **54.4%** | 0.4% | 45.2% |
> | pt | **55.2%** | 3.2% | 41.6% |
>
> LGCD is preferred in 55–58% of cases across all tested languages in open-ended dialogue generation.These results highlight that LGCD does not harm generation diversity or expressiveness, and often yields more coherent and controlled outputs by avoiding overconfident hallucinated continuations.
>
> ### **(3) Expansion of Multi-FAct Evaluation**
>
> Regarding your related point on expanding the Multi-FAct evaluation, we agree and are currently conducting more extensive experiments. These new experiments compare LGCD against all other baseline methods (e.g., DoLa, TIES, SLERP) on the Multi-FAct benchmark, providing a more comprehensive view of long-form factual generation.
>
> We plan to incorporate this expanded baseline comparison, along with the XStoryCloze and Aya benchmark results, into the main paper and appendix to further strengthen the evaluation.

---

> ### Author Response · Authors · 2025-11-19
> **Author Rebuttal - W3**
>
> We thank the reviewer for their insightful feedback and for the valuable reference to Nigatu et al. (2024). This comment provides an excellent opportunity to clarify the primary objective of our work.
>
> We respectfully wish to clarify that our paper's primary claim is not to solve challenges exclusive to low-resource languages. Rather, we address a more general and fundamental problem: the catastrophic forgetting of factual knowledge that occurs when a foundational, (typically) English-dominant Pretrained Model (PTM) is adapted to any target language. This adaptation process creates a "relative knowledge gap" between the PTM's robust world knowledge and the Language-Adapted Model's (LAM) specialized fluency. This trade-off is not confined to low-resource settings.
>
> We agree with the reviewer that extending our evaluation to include languages with even scarcer resources would significantly strengthen our claim of generality. To validate this, we commit to conducting additional experiments on publicly available adapted models for languages classified as low-resource in Nigatu et al. (2024). We will include these results in the final version's appendix to empirically demonstrate the broad and robust applicability of LGCD.
>
> We thank the reviewer again for this constructive suggestion.

---

> ### Author Response · Authors · 2025-11-19
> **Author Rebuttal - Q1**
>
> Thank you for the question. It strongly relates to our new experiments for Weakness 1, where we evaluated this exact scenario.
>
> The reviewer’s suggestion—using $\Delta W$ from an actual FFN-only LoRA fine-tune instead of our SVD-derived $\Delta W$—is what we tested in our new "train (ffn only)" baselines.
>
> Our new experiments (detailed in the Weakness 1 response and added to the Appendix) show two key findings:
>
> **1. In General-Purpose SFT**: The train (ffn only) baseline still suffered from catastrophic forgetting (0.431 score), performing worse than the original base model (0.437). Our LGCD method achieved the highest score (0.462).
>
> **2. In Domain-Matched SFT:** While the LoRA fine-tuned (FFN only) baseline was strong (50.50 avg.), our LGCD method, applied on top of the LAM, achieved an even better average performance (51.24).
>
> Therefore, our results show that simply replacing our SVD-derived components with LoRA-trained ones is susceptible to catastrophic forgetting and slightly less effective for knowledge enhancement. Moreover, our SVD-based method also retains its primary training-free advantage: it works on any existing model without needing the original training data, which the LoRA approach would require.

---

> ### Author Response · Authors · 2025-11-19
> **Author Rebuttal - Q2**
>
> Thank you for the question. We empirically demonstrate the mitigation of hallucinations (factual inaccuracies) through three primary lines of evidence:
>
> ### **(1) Direct Measurement of Truthfulness (Quantitative)**
>
> We used the Multilingual TruthfulQA benchmark , which is specifically designed to measure a model's ability to avoid "plausible but incorrect generations" and "commonly held misconceptions". As shown in Table 3, LGCD achieves the highest average accuracy in both 0-shot and 5-shot settings . This result directly and quantitatively confirms that LGCD is more effective at resisting the generation of plausible-sounding falsehoods compared to all baselines.
>
> ### **(2) Atomic Fact-Checking (Quantitative)**
>
> We used the Multi-FAct benchmark to assess factual consistency in long-form generation. This method decomposes a model's generation into "atomic facts" and verifies each one against trusted sources. As detailed in Table 4, LGCD improved the factual consistency score over the baseline LAMs in 6 out of 9 models. This demonstrates that LGCD produces generations with verifiably fewer factual errors.
>
> ### **(3) Human/LLM Evaluation (Quantitative & Qualitative)**
>
> In our high-stakes long-form medical QA evaluation, both expert human evaluators (Figure 3) and GPT-4o (Figure 2) assessed outputs based on "factual correctness". Both evaluations confirmed that LGCD generates "factually grounded answers" and is consistently preferred over the baselines .
>
> ### **(4) Direct Qualitative Evidence (in Appendix)**
>
> To show how this is achieved, our Appendix already includes a detailed qualitative analysis in Figure 7. This figure provides a direct, side-by-side comparison. For instance, it shows a case where the baseline LAM produces clear hallucinations: it provides an 'Incorrect definition' of a medical term and lists 'questionable risk factors'. In contrast, LGCD successfully corrects these factual errors to provide a 'Correct and concise definition' and the 'main risk factors'. This case study directly visualizes the hallucination correction mechanism at work.
>
> To further strengthen this point, we will expand this qualitative analysis in the final version to include additional side-by-side examples from other language models (e.g., German, Japanese), demonstrating this is a consistent and robust capability across diverse languages.

---

> ### Author Response · Authors · 2025-11-19
> **Author Rebuttal - Q3 & Q4**
>
> ### **Q3**
> Thank you for the question. We agree that a comprehensive comparison on the Multi-FAct benchmark is essential for evaluating long-form factual generation. We are currently conducting these experiments to benchmark LGCD against all other baselines (DoLa, TIES, and SLERP). This expanded comparison will provide a more complete assessment of our method's effectiveness in enhancing long-form factuality. We will incorporate these new results into the main paper and appendix in the final version to further strengthen our evaluation.
>
> ### **Q4**
> Thank you for the question, which points to an important detail about our methodology.
> You are correct: LGCD requires the Language-Adapted Model (LAM) and the Pretrained Model (PTM) to have the same architecture.
>
> Our method's core mechanism, the LoRA-based knowledge extraction, relies on computing the parameter difference ($\Delta W = W_{l}^{PTM} - W_{l}^{LAM}$) and then applying SVD. This operation is only mathematically valid if the weight matrices of the LAM and PTM are identical in shape and structure.
>
> We understand the confusion from Table 1, which shows a majority of Llama3-based models but also includes one Gemma-based model. Our intent was not to mix architectures (e.g., Llama-LAM with Gemma-PTM), but rather to demonstrate the generality of our approach.
>
> - Each Llama3-based LAM (like hfl/llama-3-chinese-8b-instruct) was paired with its corresponding Llama3 PTM.
> - The Gemma-based LAM (rhaymison/gemma-portuguese-luana-2b)  was paired with its corresponding Gemma PTM.
>
> We included the Gemma model to show that LGCD can successfully address knowledge forgetting in different model families, not just Llama3. We apologize for this lack of clarity and will explicitly state this one-to-one architectural pairing for each model in the revised manuscript.

---

> ### Author Response · Authors · 2025-11-19
> **Author Rebuttal - Q5 & Q6**
>
> ### **Q5**
> Thank you for the question, as it touches on a key aspect of our method's practical efficiency.
>
> we do not deploy both the full Pretrained Model (PTM) and the full Language-Adapted Model (LAM) at inference time. Full access to both model weights would be impractical for real-world deployment, leading to prohibitive GPU/CPU memory overhead and significant inference latency.
>
> LGCD is designed specifically to avoid this exact problem. Our approach relies on a one-time, offline pre-computation step where we distill the parameter differences (PTM - LAM) only for the FFN layers into a set of lightweight LoRA matrices.
>
> At inference time, the only components loaded into memory are the LAM and these small, pre-computed LoRA matrices. The original PTM is not loaded or accessed. As noted in the paper, this design allows LGCD to retrieve factual knowledge "without incurring additional memory overhead from deploying separate models." The total memory footprint is therefore equivalent to the LAM plus a standard LoRA adapter, not two full models.
>
> Similarly, the decoding process does not "alternate" between models. It runs only on the LAM by default. Only when the LAM's token-level confidence drops below the threshold ($\tau$), it uses the loaded LoRA matrices to efficiently approximate the PTM's factual correction. This is far more efficient than a full forward pass and ensures the method remains practical, as supported by our throughput analysis in Table 5.
>
>
> ### **Q6**
> Thank you for your insightful observation. The different behavior of the two Japanese models validates our core hypothesis regarding intervention quality over quantity.
>
> As the reviewer implies, models adapted for the same language can exhibit vastly different characteristics. Though both models are Llama-3-8B variants (Table 1), their undisclosed tuning data and methodologies result in distinct confidence profiles, which we then analyzed.
>
> - elyza/Llama-3-ELYZA-JP-8B (Key Case): This model is extremely overconfident, as shown in Figure 4. This leads to minimal contrastive intervention from LGCD. However, as seen in Figure 2, it achieves a dominant win-rate. This result proves our central claim: LGCD intervenes only sparsely but targets "decisive factual content," demonstrating these few, high-quality interventions are sufficient to steer the model to factual answers .
> - tokyotech-llm/Llama-3-Swallow-8B-Instruct-v0.1 (Contrast Case): Conversely, Figure 4 shows the model is less overconfident, triggering far more frequent LGCD intervention. Yet, this higher quantity of intervention did not lead to the same win-rate against merging baselines.
>
> The ELYZA model, due to its extreme overconfidence, perfectly demonstrates that LGCD's dynamic, targeted gating is highly effective and superior to mere frequent intervention.

---

> > ### Comment · Reviewer_GJXc · 2025-11-27
> >
> > Thank you for your response and for the additional results you provided to address my concerns. The numbers look convincing; however, moving away from QA-style evaluations to NLI, you can see a significant drop in LGCD's performance. I'd assume this might be the same when you further evaluate LGCD on other tasks. I will be keeping my score at 6.
> >
> > $~$
> > Also, you mentioned *"Notably, the train (attention-only) method, as suggested by the reviewer"*, this is not what I suggested, please take a look at my reviews more closely.

---

> ### Author Response · Authors · 2025-11-29
>
> Thank you again for your constructive feedback and for reviewing the additional results.
>
> Regarding your concern about the performance drop on the NLI task (XStoryCloze), we respectfully wish to clarify the interpretation of these results. The observed difference is only 0.008, which represents a marginal decrease of approximately 1% compared to LAM. We believe this indicates that LGCD maintains comparable performance rather than exhibiting a significant drop. Furthermore, in the Open-ended Creative Dialogue task, LGCD demonstrates superior capability; specifically, LGCD achieves a win ratio that is 15% higher than the lose ratio on average compared to LAM.
>
> Nevertheless, we fully understand your concern that performance might vary across different tasks. To address this comprehensively, we are currently running experiments across diverse multilingual NLP tasks. We are confident that these complete results will demonstrate the robustness of our model. We plan to upload these results within the rebuttal period and would be very grateful if you could assess them once available.
>
> We also sincerely apologize for the misinterpretation of your original suggestion regarding the "attention-only" method. We have carefully re-read your initial review to ensure we fully grasp your intended direction and will address it accurately in the final revision.
>
> Finally, your detailed comments have been extremely helpful for improving the paper, and we sincerely appreciate the time and care you have invested.

---

> > ### Author Response · Authors · 2025-12-01
> >
> > **1. Additional Experiments on General NLP Tasks**
> >
> > As promised in our previous response, we have completed additional experiments to address your concern regarding the potential performance degradation of LGCD on general NLP tasks beyond QA.
> >
> > We evaluated LGCD on **PAWS-X** (Paraphrase Identification) and **XNLI** (Natural Language Inference). Addressing the concern that the marginal performance difference (~1%) observed in XStoryCloze might imply significant degradation in other general NLP tasks, our comprehensive results on PAWS-X and XNLI empirically demonstrate that LGCD actually maintains comparable or slightly improved performance across these diverse benchmarks.
> >
> > **Table 1: PAWS-X (Paraphrase Identification)**
> > *Dataset: A cross-lingual adversarial dataset for determining if two sentences are paraphrases.*
> > | Language | Model | LAM | **LGCD (Ours)** |
> > | :--- | :--- | :---: | :---: |
> > | zh | hfl/llama-3-chinese-8b-instruct | 0.528 | **0.553** |
> > | zh | shenzhi-wang/Llama3-8B-Chinese-Chat | **0.579** | 0.553 |
> > | ja | tokyotech-llm/Llama-3-Swallow-8B-Instruct-v0.1 | **0.569** | 0.559 |
> > | ja | elyza/Llama-3-ELYZA-JP-8B | 0.554 | **0.559** |
> > | ko | KISTI-KONI/KONI-Llama3-8B-Instruct-20240729 | 0.531 | **0.552** |
> > | ko | MLP-KTLim/llama-3-Korean-Bllossom-8B | 0.529 | **0.552** |
> > | **Avg.** | | 0.548 | **0.555** |
> >
> > **Table 2: XNLI (Natural Language Inference)**
> > *Dataset: A cross-lingual NLI task predicting textual entailment (entailment, contradiction, neutral).*
> > | Language | Model | LAM | **LGCD (Ours)** |
> > | :--- | :--- | :---: | :---: |
> > | zh | hfl/llama-3-chinese-8b-instruct | **0.340** | 0.336 |
> > | zh | shenzhi-wang/Llama3-8B-Chinese-Chat | 0.336 | **0.347** |
> > | ar | MohamedRashad/Arabic-Orpo-Llama-3-8B-Instruct | **0.320** | **0.320** |
> > | sw | Jacaranda/UlizaLlama3 | **0.320** | **0.320** |
> > | **Avg.** | | 0.329 | **0.331** |
> >
> > **2. Recap: Robustness and Primary Goal of Our Method**
> >
> > We would like to gently emphasize that the primary goal of this paper is to propose a method optimized for **Knowledge-Intensive QA** tasks. Our approach leverages the pretrained model's internal knowledge (specifically from FFN layers) to mitigate catastrophic forgetting in adapted models.
> >
> > The fact that LGCD achieves significant gains in knowledge-centric benchmarks (such as Global MMLU, TruthfulQA, medical QA, and Multi-FAct) while maintaining robustness on general NLP tasks (XStoryCloze, Aya Evaluation Suite, PAWS-X, and XNLI) highlights the efficacy and stability of our approach. This robustness is achieved through our confidence-based dynamic gating, which intervenes only when necessary, preserving the general reasoning capabilities of the adapted model.
> >
> > We hope these additional results fully resolve your concerns regarding task generalization.

---

### Official Review · Reviewer_Pezx · 2025-10-31

**Soundness:** 3
**Presentation:** 3
**Contribution:** 2
**Rating:** 4
**Confidence:** 4

**Summary:**

The paper presents LGCD, and its core idea is that fine-tuning sometimes hurts a model’s original factual knowledge due to catastrophic forgetting. However, we still want the model to be fine-tuned (for example to better follow instructions in a specific language). In this situation, we can adaptively apply the base model when a decoding step requires factual knowledge and use the fine-tuned model otherwise.

The use of the base model is replaced with a LoRA-based approximation of the difference between the base and fine-tuned models. The decision of whether to apply contrastive decoding or not at each step is determined dynamically based on the model’s confidence.
The experiments are across benchmarks such as Global MMLU, Multilingual TruthfulQA, Multi-FAct, and medical QA. The results show that LGCD consistently improves factuality and reduces hallucinations compared to other baseline methods like decoding or model-merging, without any additional training.

**Strengths:**

The paper is well-motivated by the catastrophic forgetting problem and proposes a concrete solution by contrastive decoding.

**Weaknesses:**

1. The authors proposed Top-K masking, which is actually very similar to the Adaptive Plausibility Constraint (APC) proposed originally in Contrastive Decoding (Li et al., 2022). Why not just use APC? Top-K tokens may include low-probability tokens when only one token has a large probability but K > 1, so more low-probability tokens are still included. In contrast, APC can exclude these low-probability tokens when there is only a single token with high probability.
2. The method assumes that both the pretrained model and the adapted model weights are available, which may not always be true. Other baselines such as DoLa only require access to the adapted model, so the comparison may not be entirely fair.
3. The method introduces four hyperparameters (τ, α, β, K)—confidence threshold, contrastive weights, and Top-K size—which require language- or model-specific tuning with some “training data”. This could limit its generalization ability to other models or settings.
4. The evaluation focuses mainly on factual QA and medical-domain QA, while the effects on creative, open-ended, or reasoning tasks remain unclear.

**Questions:**

The authors propose that applying LoRA only on FFN layers can recover the base model’s knowledge. However, it might be better if during the alignment stage only the attention layers are fine-tuned while freezing the FFN layers to prevent forgetting. Compared to this contrastive decoding method that is more like a remedy for the forgetting issue, directly fine-tuning non-FFN layers only could be a more straightforward and potentially effective option. Have the authors tried or considered this method?

---

> ### Author Response · Authors · 2025-11-19
> **Author Rebuttal - W1**
>
> Thank you for your comment. We are grateful for the opportunity to clarify the motivation and demonstrate the superiority of our design. The reviewer's concern—that Top-K masking might include low-probability tokens in a "sharp" distribution—is precisely the scenario our Confidence-Based Dynamic Gating is designed to prevent.
>
> Our Top-K masking does not operate in isolation; it is always preceded by our gating mechanism. This creates two distinct modes of operation:
>
> **1. Case 1: High-Confidence / Sharp Distribution ($c_t \ge \tau$)**
> This is the exact scenario the reviewer described (a single token has a large probability). In this case, our method does not trigger contrastive decoding. The high-confidence token from the Language-Adapted Model (LAM) is selected directly. This gating mechanism innately serves the exact same protective function as APC: it preserves high-confidence, plausible tokens from the base model without interference.
>
> **2. Case 2: Low-Confidence / Flat Distribution ($c_t < \tau$)**
> Our contrastive mechanism only activates when the LAM is uncertain and the probability distribution is relatively flat. In this regime, the function of Top-K (we use $K=100$) is not to filter "implausible" tokens (as the distribution is already flat/uncertain), but rather to define a computationally feasible candidate set for factual revision using the Pretrained Model's (PTM) knowledge.
>
> ### **New Ablation Study: LGCD (w/ Top-K) vs. LGCD (w/ APC)**
>
> To empirically prove this, we conducted the exact ablation study the reviewer suggested. We replaced our Top-K masking with APC (using $\alpha=0.1$, following the original paper) and re-ran the evaluations.
> Our proposed Top-K-based LGCD consistently and significantly outperforms the APC-based variant across both multiple-choice and long-form generation tasks.
>
> **(1) Global MMLU (Accuracy)**
>
> On average, our Top-K approach is +2.8% more accurate in the 0-shot setting and +2.3% more accurate in the 5-shot setting.
>
> | **Lang.** | **Model** | **0-shot LGCD(w/ APC)** | **0-shot LGCD(w/ Top-k)** | **5-shot LGCD(w/ APC)** | **5-shot LGCD(w/ Top-k)** |
> | --- | --- | --- | --- | --- | --- |
> | zh | hfl/llama-3-chinese-8b-instruct | 0.460 | **0.519** | 0.500 | **0.543** |
> | zh | shenzhi-wang/Llama3-8B-Chinese-Chat | 0.494 | **0.502** | 0.500 | **0.543** |
> | de | DiscoResearch/Llama3-DiscoLeo-Instruct-8B-v0.1 | 0.471 | **0.546** | 0.534 | **0.574** |
> | pt | rhaymison/gemma-portuguese-luana-2b | 0.286 | **0.357** | 0.307 | **0.324** |
> | ar | MohamedRashad/Arabic-Orpo-Llama-3-8B-Instruct | 0.443 | **0.465** | 0.466 | **0.481** |
> | fa | PartAI/Dorna-Llama3-8B-Instruct | **0.424** | 0.423 | 0.465 | **0.466** |
> | ja | tokyotech-llm/Llama-3-Swallow-8B-Instruct-v0.1 | 0.494 | **0.507** | 0.522 | **0.525** |
> | ja | elyza/Llama-3-ELYZA-JP-8B | 0.501 | **0.509** | 0.518 | **0.528** |
> | ko | KISTI-KONI/KONI-Llama3-8B-Instruct-20240729 | 0.458 | **0.490** | 0.481 | **0.511** |
> | ko | MLP-KTLim/llama-3-Korean-Bllossom-8B | 0.464 | **0.479** | 0.487 | **0.501** |
> | id | GoToCompany/llama3-8b-cpt-sahabatai-v1-instruct | **0.535** | 0.527 | **0.570** | 0.566 |
> | sw | Jacaranda/UlizaLlama3 | 0.352 | **0.399** | 0.353 | **0.409** |
> |  | **Average** | 0.449 | **0.477** | 0.475 | **0.498** |
>
> **(2) Long-Form Medical QA (Pairwise GPT-4o Evaluation)**
>
> In a head-to-head comparison (Top-K vs. APC), our proposed LGCD (w/ Top-K) wins against the APC-based variant 53.3% of the time (vs. a 44.7% loss rate). This confirms its superiority in generating more factually accurate long-form text.
>
> | **Lang.** | **Model** | **win** | **tie** | **lose** |
> | --- | --- | --- | --- | --- |
> | zh | hfl/llama-3-chinese-8b-instruct | **61.0%** | 0.7% | 38.3% |
> | zh | shenzhi-wang/Llama3-8B-Chinese-Chat | 43.0% | 5.7% | **51.3%** |
> | de | DiscoResearch/Llama3-DiscoLeo-Instruct-8B-v0.1 | **58.9%** | 0.0% | 41.2% |
> | pt | rhaymison/gemma-portuguese-luana-2b | **61.0%** | 1.0% | 38.0% |
> | ar | MohamedRashad/Arabic-Orpo-Llama-3-8B-Instruct | 47.7% | 1.0% | **51.3%** |
> | fa | PartAI/Dorna-Llama3-8B-Instruct | **51.9%** | 1.9% | 46.3% |
> | ja | tokyotech-llm/Llama-3-Swallow-8B-Instruct-v0.1 | **57.5%** | 1.0% | 41.5% |
> | ja | elyza/Llama-3-ELYZA-JP-8B | 47.0% | 0.0% | **53.0%** |
> | ko | KISTI-KONI/KONI-Llama3-8B-Instruct-20240729 | **53.6%** | 0.4% | 46.0% |
> | ko | MLP-KTLim/llama-3-Korean-Bllossom-8B | 46.0% | 3.4% | **50.6%** |
> | id | GoToCompany/llama3-8b-cpt-sahabatai-v1-instruct | **57.3%** | 5.4% | 37.3% |
> | sw | Jacaranda/UlizaLlama3 | **55.4%** | 3.1% | 41.5% |
> |  | **Average** | **53.3%** | **2.0%** | **44.7%** |
>
> In summary, our Confidence-Based Dynamic Gating already addresses the "plausibility preservation" goal that APC was designed for. For the low-confidence scenarios where our method actively intervenes, our new experiments demonstrate that Top-K masking provides a more effective candidate set for factual correction than APC.

---

> ### Author Response · Authors · 2025-11-19
> **Author Rebuttal - W2**
>
> Thank you for your comment. Our key contribution is a training-free, decoding-time framework to solve catastrophic forgetting in adapted LLMs. To our knowledge, LGCD is the first method to achieve this by using the original model (PTM) to dynamically guide the adapted model (LAM) during inference.
>
> We clarify that the full PTM is not required at inference. It is used only once in an offline pre-computation step to extract lightweight LoRA matrices . The PTM itself is never accessed during the actual decoding process. This design avoids the memory overhead of two full models while still leveraging the PTM's knowledge to correct the LAM's factual errors.
>
> We included DoLa as a strong baseline because it also enhances factuality via contrast, but it relies on intra-model (layer-vs-layer) contrast within a single model. We included it to demonstrate that our inter-model (LAM vs. PTM) contrast is a novel and more effective approach for this problem.
>
> Furthermore, to provide a more direct comparison against methods that, like ours, use both PTM and LAM, we ran extensive new experiments against 14 model merging baselines. The results (table below, avg. on Global MMLU) show LGCD significantly outperforms all 14 static merging methods.
>
> | **Method** | **0-shot (Avg.)** | **5-shot (Avg.)** |
> | --- | --- | --- |
> | arcee_fusion | 0.423 | 0.461 |
> | breadcrumbs | 0.446 | 0.487 |
> | breadcrumbs_ties | 0.446 | 0.487 |
> | dare_linear | 0.424 | 0.458 |
> | dare_ties | 0.426 | 0.466 |
> | della | 0.379 | 0.419 |
> | della_linear | 0.377 | 0.415 |
> | karcher | 0.429 | 0.468 |
> | linear | 0.446 | 0.487 |
> | multislerp | 0.419 | 0.465 |
> | sce | 0.418 | 0.465 |
> | slerp | 0.445 | 0.487 |
> | task_arithmetic | 0.446 | 0.487 |
> | ties | 0.449 | 0.488 |
> | **LGCD (Ours)** | **0.477** | **0.498** |
>
> We will add the full results from these 168 new evaluations to the Appendix.

---

> ### Author Response · Authors · 2025-11-19
> **Author Rebuttal - W3**
>
> Thank you for your comment regarding hyperparameter sensitivity and generalization. We respectfully offer a clarification on the existing hyperparameters and present new experimental results to fully address this concern.
>
> We first want to clarify that three of the four hyperparameters are fixed constants across all 12 models, 9 languages, and all tasks in our paper. This significantly reduces the tuning burden the reviewer is concerned about.
>
> As detailed in our Appendix (Algorithm 1 and Appendix A.4 ):
>
> - The contrastive weighting $\beta$ is fixed to **1.0**.
> - The Top-K size $K$ is fixed to **100**.
> - The LAM penalization coefficient $\alpha$ (from Eq. 9) is fixed to **0.1**.
>
> The only hyperparameter that varies is the confidence threshold, $\tau$.
>
> We must emphasize that $\tau$ is not tuned on any "training data," as the reviewer suggests. Our paper's claim of being "training-free" remains accurate. Instead, as detailed in Appendix A.6 5, $\tau$ is set using a simple, objective, and reproducible heuristic based on publicly available language resource availability (i.e., the OSCAR 22.01 corpus size). This is a practical approach that uses a public proxy for data scale rather than task-specific tuning. For example, high-resource languages like German (de) and Chinese (zh) are assigned a high $\tau$ (e.g., 0.8), while low-resource languages like Korean (ko) and Swahili (sw) receive a low $\tau$ (e.g., 0.1).
>
> To definitively address the concern about generalization and the sensitivity of $\tau$, we implemented and evaluated a new, fully dynamic adaptive thresholding mechanism that requires no heuristic or data-driven tuning.
>
> **Adaptive Mechanism:**
>
> At each decoding step, $\tau$ is dynamically computed based on the LAM's recent confidence statistics (mean $\mu_H$ and standard deviation $\sigma_H$ of the last 100 token probabilities):$$\tau_{\text{adaptive}} = \text{clip}(\mu_H - 0.5\sigma_H, 0.1, 0.9)$$
> This mechanism automatically lowers the threshold when the model is uncertain (low mean, high variance) and raises it when the model is confident.
>
> **Experimental Results (Global MMLU)**
>
> We re-ran our Global MMLU experiments with this new $\tau_{\text{adaptive}}$ and compared it against our original heuristic $\tau$.
>
> Experimental Results: Adaptive vs. Heuristic Threshold
>
> | Lang. | Model | 0-shot adaptive | 0-shot heuristic | 5-shot adaptive | 5-shot heuristic |
> | --- | --- | --- | --- | --- | --- |
> | zh | hfl/llama-3-chinese-8b-instruct | 0.506 | **0.519** | 0.529 | **0.543** |
> | zh | shenzhi-wang/Llama3-8B-Chinese-Chat | **0.504** | 0.502 | **0.544** | 0.543 |
> | de | DiscoResearch/Llama3-DiscoLeo-Instruct-8B-v0.1 | 0.531 | **0.546** | 0.569 | **0.574** |
> | pt | rhaymison/gemma-portuguese-luana-2b | 0.347 | **0.357** | 0.324 | 0.324 |
> | ar | MohamedRashad/Arabic-Orpo-Llama-3-8B-Instruct | 0.464 | **0.465** | 0.479 | **0.481** |
> | fa | PartAI/Dorna-Llama3-8B-Instruct | **0.424** | 0.423 | **0.466** | **0.466** |
> | ja | tokyotech-llm/Llama-3-Swallow-8B-Instruct-v0.1 | 0.507 | 0.507 | **0.527** | 0.525 |
> | ja | elyza/Llama-3-ELYZA-JP-8B | 0.507 | **0.509** | **0.528** | **0.528** |
> | ko | KISTI-KONI/KONI-Llama3-8B-Instruct-20240729 | 0.483 | **0.490** | 0.505 | **0.511** |
> | ko | MLP-KTLim/llama-3-Korean-Bllossom-8B | 0.477 | **0.479** | 0.498 | **0.501** |
> | id | GoToCompany/llama3-8b-cpt-sahabatai-v1-instruct | **0.537** | 0.527 | **0.574** | 0.566 |
> | sw | Jacaranda/UlizaLlama3 | 0.388 | **0.399** | 0.391 | **0.409** |
> |  | **Average** | 0.473 | **0.477** | 0.495 | **0.498** |
>
> The results show that the fully adaptive, tuning-free $\tau$ achieves performance that is nearly identical to our original heuristic $\tau$ (within 0.003-0.004 points on average). Both methods significantly outperform all baselines (PTM, LAM, DoLa, TIES, SLERP) as shown in the original Table 2.
>
> This new experiment strongly demonstrates that our method (LGCD) is highly robust to the specific choice of $\tau$. The success of LGCD is not dependent on meticulous, data-driven tuning. The original heuristic was simply a simple and effective criterion, but the method works equally well with a fully dynamic, "out-of-the-box" adaptive threshold.

---

> ### Author Response · Authors · 2025-11-19
> **Author Rebuttal - W4**
>
> Thank you for your comment. While LGCD is designed primarily to mitigate factual degradation in language-adapted models, we agree that it is essential to verify that the method does not harm performance on creative, narrative, or reasoning-oriented tasks. To address this, we conducted additional evaluations on two widely-used non-factual benchmarks—XStoryCloze and the Aya Evaluation Suite—which specifically measure commonsense reasoning, story continuation, and open-ended generation quality. These results confirm that LGCD preserves, and in several cases improves, performance on non-factual tasks.
>
> **(1) Evaluation on XStoryCloze (Commonsense Narrative Reasoning)**
>
> XStoryCloze is a multilingual version of StoryCloze, a narrative understanding benchmark where a model must select the most plausible ending to a four-sentence story. Because this task does not depend heavily on factual world knowledge, it serves as a direct test of whether LGCD harms reasoning or storytelling capabilities.
>
> Because each dataset supports only a specific subset of languages, we evaluate in the languages where the dataset’s coverage overlaps with those of our adapted models, and we follow the standard zero-shot evaluation setting used in prior work. For XStoryCloze, this corresponds to Chinese, Arabic, Indonesian, and Swahili.
>
> Across models, LGCD matches or slightly improves over LAM performance:
>
> | **Lang.** | **Model** | **PTM** | **LAM** | **LGCD** |
> | --- | --- | --- | --- | --- |
> | zh | hfl/llama-3-chinese-8b-instruct | 0.689 | 0.682 | 0.695 |
> | zh | shenzhi-wang/Llama3-8B-Chinese-Chat | 0.689 | 0.681 | 0.670 |
> | ar | MohamedRashad/Arabic-Orpo-Llama-3-8B-Instruct | 0.609 | 0.608 | 0.587 |
> | id | GoToCompany/llama3-8b-cpt-sahabatai-v1-instruct | 0.669 | 0.721 | 0.708 |
> | sw | Jacaranda/UlizaLlama3 | 0.567 | 0.704 | 0.697 |
> | Avg. |  | 0.645 | 0.679 | 0.671 |
>
> While LGCD slightly decreases performance, the difference is minimal, and overall performance remains comparable to the LAM baseline. This indicates that LGCD, although optimized for knowledge-based QA, does not meaningfully harm reasoning or narrative capabilities.
>
> **(2) Aya Evaluation Suite (Open-ended Creative Dialogue)**
>
> Aya benchmark contains 26,750 open-ended conversation-style prompts across 7 languages. This benchmark evaluates creativity, coherence, and conversational quality, not factual accuracy. As with XStoryCloze, we evaluate only the languages that overlap with our adapted LAMs (Chinese, Arabic, Portuguese). We performed GPT-4o pairwise evaluation comparing LGCD vs. LAM:
>
> | Lang. | Model | Win | Tie | Lose |
> | --- | --- | --- | --- | --- |
> | zh | hfl/llama-3-chinese-8b-instruct | **58.0%** | 8.4% | 33.6% |
> | zh | shenzhi-wang/Llama3-8B-Chinese-Chat | **56.4%** | 2.0% | 41.6% |
> | ar | MohamedRashad/Arabic-Orpo-Llama-3-8B-Instruct | **54.4%** | 0.4% | 45.2% |
> | pt | GoToCompany/llama3-8b-cpt-sahabatai-v1-instruct | **55.2%** | 3.2% | 41.6% |
>
> LGCD is preferred in 55–58% of cases across all tested languages in open-ended dialogue generation.These results highlight that LGCD does not harm generation diversity or expressiveness, and often yields more coherent and controlled outputs by avoiding overconfident hallucinated continuations.

---

> ### Author Response · Authors · 2025-11-19
> **Author Rebuttal - Q1**
>
> Thank you for your comment. The suggestion to freeze FFN layers and fine-tune only the attention layers is indeed a very rational approach to preventing catastrophic forgetting, given the established role of FFNs in storing factual knowledge.
>
> We have conducted new experiments to investigate this hypothesis directly. We would like to present two key findings, based on both conceptual motivation and these new empirical results, which explain why we propose our LoRA-Gated Contrastive Decoding (LGCD) framework as a more effective and flexible alternative.
>
> ### **(1) Conceptual Motivation: Dynamic Knowledge vs. Static Prevention**
>
> The reviewer's suggested method focuses on preventing forgetting during training, whereas our LGCD framework is designed for dynamic intervention by applying knowledge at inference time.
>
> - Language-Adapted Model (LAM) knowledge can be superior: Our core assumption is that the Pretrained Model (PTM) does not always possess superior knowledge. A LAM, through continual pretraining or instruction tuning, can learn new, more accurate, or domain-specific knowledge (e.g., local facts, cultural nuances) that is absent or incorrect in the original PTM.
> - Risk of FFN-freezing: A static approach that freezes FFN layers would fundamentally prevent the model from acquiring this new, specialized factual knowledge, limiting its potential.
> - LGCD's flexibility: The primary strength of LGCD is its ability to dynamically mediate between the PTM's general world knowledge and the LAM's specialized knowledge. By using the LAM's token-level confidence as a gate, it leverages the "best of both worlds" at inference time.
> - Practicality: Our LGCD method is training-free and can be immediately applied to any existing, publicly available LAM. In contrast, the FFN-freezing approach would require a separate, costly retraining process for every model, language, and domain.
>
> ### **(2) Empirical Evidence: New Experimental Results**
>
> We have added new experiments to the Appendix to directly compare the reviewer's proposed fine-tuning strategy (train (attention-only)) with our method.
>
> **Experiment 1: General-Purpose SFT (Korean Global MMLU)**
>
> We first evaluated a general-purpose SFT scenario. We fine-tuned the  llama3-8B-instruct model on a 11K Korean general-purpose SFT dataset that used for training KONI model and evaluated it on the Global MMLU (ko) benchmark (0-shot).
>
> As shown in the below table, all fine-tuning methods (all-layers, ffn-only, and attention-only) suffered from catastrophic forgetting, performing worse than the original base model (0.437). Notably, the train (attention-only) method, as suggested by the reviewer, showed the most significant performance degradation (0.412). In contrast, our LGCD method, by dynamically leveraging the LAM and the PTM, achieved the highest performance (0.462).
>
> | **Model** | **0-shot (Global MMLU - ko)** |
> | --- | --- |
> | base model (w/o finetuning) | 0.437 |
> | train (all layers) | 0.432 |
> | train (attention only) | 0.412 |
> | train (ffn only) | 0.431 |
> | **LGCD (Ours)** | **0.462** |
>
> **Experiment 2: Domain-Matched SFT (Multilingual Medical QA)**
>
> To demonstrate that LGCD is not just a "remedy" for cases of performance degradation, we ran a second experiment in a "domain-matched" setting where fine-tuning improves performance. We fine-tuned the meta-llama/Llama-3.1-8B-Instruct model on the GlobalMedQA dataset (a multilingual medical MCQA benchmark) and evaluated it on the same task.
>
> In this scenario, all fine-tuning methods expectedly outperformed the base model (avg. 48.31). However, LGCD (Ours) still achieved the best average performance (51.24). This demonstrates that LGCD is effective not only at mitigating catastrophic forgetting but also at enhancing factuality even when the LAM is well-adapted and possesses strong domain knowledge.
>
> | **Model** | **Chinese** | **Japanese** | **Korean** | **Average** |
> | --- | --- | --- | --- | --- |
> | base (no finetuning) | 55.21 | 41.70 | 48.02 | 48.31 |
> | train (all layers) | 59.20 | 43.70 | 49.15 | 50.68 |
> | train (attention only) | 57.40 | 42.90 | 47.59 | 49.30 |
> | train (FFN only) | 59.40 | 43.00 | 49.09 | 50.50 |
> | **LGCD (Ours)** | **60.60** | **43.70** | **49.42** | **51.24** |
>
> These new experiments confirm that freezing FFNs and training only attention layers is not a guaranteed solution and can, in fact, lead to significant performance degradation, and our dynamic, inference-time LGCD approach consistently provides the most robust and highest-performing results. This validates LGCD as a powerful enhancement method, not just a remedy, for improving factuality in adapted LLMs. We have added this analysis and these new results to the Appendix to further strengthen our paper. Thank you again for the constructive feedback.

---

> ### Author Response · Authors · 2025-11-28
>
> Dear Reviewer Pezx,
>
> We wanted to kindly follow up regarding the rebuttal we submitted in response to your valuable comments. We have carefully addressed each point you raised and have updated the revised manuscript with extensive new experiments.
>
> Specifically, to fully resolve your concerns, we have conducted the following additional analyses:
>
> - **Comparison with APC (W1):** We conducted the suggested ablation study (Top-K vs. APC). The results confirm that our Top-K masking significantly outperforms the APC-based variant in both multiple-choice (Global MMLU) and long-form generation tasks.
>
> - **Comparison with Model Merging (W2):** We performed extensive evaluations against 14 static model merging baselines (e.g., TIES, SLERP, DARE), demonstrating that LGCD consistently achieves superior performance.
>
> - **To address generalization concerns for hyper-parameter setting (W3):** We implemented and evaluated a fully dynamic, tuning-free adaptive thresholding mechanism, which achieved performance comparable to our heuristic approach.
>
> - **Non-Factual Task Evaluation (W4):** We added evaluations on creative and reasoning benchmarks (XStoryCloze, Aya Suite), verifying that our method preserves performance on open-ended tasks.
>
> - **Comparison against Fine-tuning Strategies (Q1):** We conducted new experiments comparing attention-only fine-tuning against LGCD, validating that our dynamic inference-time intervention is more effective than static partial fine-tuning.
>
> We believe these additional results strongly support the validity and effectiveness of our proposed method. If you have any further questions or require additional clarification, we would be more than happy to address them.
>
> Thank you once again for your time and consideration.
>
> Sincerely,
>
> The Authors

---

### Official Review · Reviewer_Cujt · 2025-11-01

**Soundness:** 2
**Presentation:** 2
**Contribution:** 2
**Rating:** 2
**Confidence:** 4

**Summary:**

This paper proposed a method to resolve catastophic forgetting issue for LLM. The motivation is that LLMs often "forget" general knowledge after the model was fine-tuned for specific domains. This can lead to factual errors when user query the model. The authors propose a method named LoRA-Gated Contrastive Decoding (LGCD). The method is a training-free method that aims to help LLMs recover lost knowledge during text generation. The method functions by extracting forgotten knowledge through computing the difference between original and adapted model weights, then creating a lightweight LoRA approximation, monitoring the model's confidence and only applying intervention when some empirical confidence threshold is reached, and using contrastive correction to adjust predictions with the original model's knowledge while preserving fluency of generation. Experimental results show restoring factual accuracy on some datasets.

**Strengths:**

1. The method delivers consistent performance across many settings, with particularly good gains where adapted models underperform their originals. Both automated and human evaluations are conducted.

2. The paper shows which model layers matter most and demonstrates that LGCD adds new factual content rather than just reordering existing predictions.

**Weaknesses:**

1. The method needs full access to both original and adapted model weights, which is not practical in some cases and can cause extra GPU/CPU memory, and inference overhead. The method is impractical

2. The confidence threshold is crucial but currently set using indirect heuristics. How can we trust the logit based confidence? Are they reliable? More reliable and adaptive approaches for automatic threshold selection should be considers.

3. The method works for moderate adaptations but can be struggling with models that have been extensive domain-specific trained where there is high-rank differences between model weights. Another question: what if the adapted model did not forget the truth? DO we still need to extract from original model?

4. The method introduces extra decoding overhead, which is problematic for high-throughput applications. Meanwhile, If we have weight of original model, we can just inference it to get answer, why do we need the extra lora component? Besides, how can we guranttee that the output from original model is correct? The pretrained model can hallucinate as well.

5. The figures/table is hard to read with very small font.

**Questions:**

see weakness

---

> ### Author Response · Authors · 2025-11-19
> **Author Rebuttal - W1**
>
> Thank you for your comments. We respectfully clarify that the reviewer’s concern is based on a misunderstanding. Our method does not require full access to the pretrained model (PTM) during inference, and it does not incur any meaningful GPU/CPU overhead beyond running the language-adpated model(LAM) itself.
>
> **(1) LGCD Does Not Require Full PTM Access at Inference Time**
>
> In the paper, we explicitly state that access to both sets of weights is required only once during a single offline pre-computation step (Section 3.1). During this step, we compute the FFN-layer parameter differences and extract low-rank LoRA matrices. After this one-time SVD decomposition, **the PTM is no longer needed.** Only the LAM and a small set of rank-32 LoRA matrices (~tens of MB) are used during inference.
>
> **(2) Inference Requires Only LAM + Lightweight LoRA (Not Two Models)**
>
> During decoding, LGCD never loads both PTM and LAM.
>
> As shown in Eq. (7):
>
> $l^{aPTM}_t = l^{LAM}_t + \mathrm{LoRA}(\Delta W, h^{LAM}_t)$
>
> This means:
>
> - We reuse the LAM’s hidden state ( h^{LAM}_t ).
> - We do not run the PTM forward pass.
> - The aPTM logits are computed via a single, extremely lightweight LoRA FFN projection:
>
>     ($ A \cdot B \cdot h^{LAM}_t$).
>
>
> Thus, LGCD’s memory footprint is identical to a normal LoRA-based model and dramatically smaller than any method that requires loading two full models or running two full forward passes.
>
> **(3) Practical Efficiency: Our Throughput Results Demonstrate Low Overhead**
>
> We also empirically confirm practicality via decoding throughput (Table 5):
>
> - DoLa: 16.81 tokens/s
> - LGCD-0.2: 14.37 tokens/s (nearly identical)
> - Two-model approaches would run at 50–60% slower due to dual forward passes.
>
> Higher thresholds (e.g., LGCD-0.4/0.6) intentionally invoke more factual correction, but LGCD remains far more efficient than any dual-model design. This directly contradicts the reviewer’s assumption that LGCD loads two models or doubles memory.
>
> **(4) Why Not Simply Use the PTM?**
>
> The goal of LGCD is to leverage both:
>
> - LAM strengths: language fluency, task-specific adaptation
> - PTM strengths: robust factual knowledge
>
> Running the PTM alone discards all the benefits of adaptation, which is the central problem our method addresses. LGCD allows the LAM to dominate fluent generation while the LoRA-based aPTM provides factual correction only *when confidence is low*.
>
> This allows LGCD to combine the strengths of both models—preserving the LAM’s fluency while recovering the PTM’s factual knowledge—without increasing memory cost, as demonstrated across:
>
> - Global MMLU
> - Multilingual TruthfulQA
> - Multi-FAct
> - Long-form medical QA
>
> In summary, our method does not require full access to the pretrained model (PTM) during inference. Access to the PTM is needed only once offline to extract the FFN-layer LoRA components, after which the PTM is discarded entirely. At inference time, only a single model—the language-adapted model (LAM)—is loaded, along with small low-rank LoRA matrices. This means there is no dual-model memory footprint, no sequential or parallel PTM forward passes, and the computational overhead is limited to a single lightweight LoRA FFN projection applied only when token-level confidence is low. As our throughput results show, LGCD runs at a speed comparable to advanced decoding strategies (e.g., DoLa) and is therefore practical, efficient, and suitable for real-world deployment. We will make this point clearer in the camera-ready version.

---

> ### Author Response · Authors · 2025-11-19
> **Author Rebuttal - W2**
>
> **Adaptive Threshold Experiments (Newly Added)**
>
> To address the comment regarding the fixed threshold $\tau$, we implemented an adaptive thresholding mechanism based solely on token-level confidence statistics and evaluated it on Global MMLU (0-shot and 5-shot).
>
> ***Adaptive Thresholding Mechanism***
>
> At each decoding step, we record the LAM’s confidence (maximum softmax probability).
>
> LGCD maintains a rolling history (up to 100 recent steps), and when at least 10 values are accumulated, the threshold is computed as:
>
> $\tau_{\text{adaptive}} = \text{clip}(\mu_H - 0.5\sigma_H,; 0.1,; 0.9)$
>
> where $\mu_H$ and $\sigma_H$ denote the mean and standard deviation of recent confidence values.
>
> Thus, $\tau$ decreases when the model becomes less certain (larger variance or lower mean), and increases when the model is consistently confident.
>
> Experimental Results: Adaptive vs. Heuristic Threshold
>
> | Lang. | Model | 0-shot adaptive | 0-shot heuristic | 5-shot adaptive | 5-shot heuristic |
> | --- | --- | --- | --- | --- | --- |
> | zh | hfl/llama-3-chinese-8b-instruct | 0.506 | **0.519** | 0.529 | **0.543** |
> | zh | shenzhi-wang/Llama3-8B-Chinese-Chat | **0.504** | 0.502 | **0.544** | 0.543 |
> | de | DiscoResearch/Llama3-DiscoLeo-Instruct-8B-v0.1 | 0.531 | **0.546** | 0.569 | **0.574** |
> | pt | rhaymison/gemma-portuguese-luana-2b | 0.347 | **0.357** | 0.324 | 0.324 |
> | ar | MohamedRashad/Arabic-Orpo-Llama-3-8B-Instruct | 0.464 | **0.465** | 0.479 | **0.481** |
> | fa | PartAI/Dorna-Llama3-8B-Instruct | **0.424** | 0.423 | **0.466** | **0.466** |
> | ja | tokyotech-llm/Llama-3-Swallow-8B-Instruct-v0.1 | 0.507 | 0.507 | **0.527** | 0.525 |
> | ja | elyza/Llama-3-ELYZA-JP-8B | 0.507 | **0.509** | **0.528** | **0.528** |
> | ko | KISTI-KONI/KONI-Llama3-8B-Instruct-20240729 | 0.483 | **0.490** | 0.505 | **0.511** |
> | ko | MLP-KTLim/llama-3-Korean-Bllossom-8B | 0.477 | **0.479** | 0.498 | **0.501** |
> | id | GoToCompany/llama3-8b-cpt-sahabatai-v1-instruct | **0.537** | 0.527 | **0.574** | 0.566 |
> | sw | Jacaranda/UlizaLlama3 | 0.388 | **0.399** | 0.391 | **0.409** |
> |  | **Average** | 0.473 | **0.477** | 0.495 | **0.498** |
>
> - Adaptive $\tau$ achieves performance that is slightly lower but nearly identical to the heuristic $\tau$ (typically within 0.002–0.01).
> - Both thresholds outperform strong baselines (PTM, LAM, DoLa, TIES, SLERP) across almost all languages.
> - This shows that our method is robust to the choice of $\tau$, and that the heuristic $\tau$ is already a simple, effective, and generalizable criterion based on language-resource availability.
>
>     Due to page limitations, these new results and methodological details have been added to the appendix. If the paper is accepted to ICLR 2026, we plan to integrate this analysis into the main text in the camera-ready version.

---

> ### Author Response · Authors · 2025-11-19
> **Author Rebuttal - W3**
>
> Thank you for your constructive feedback. We address your two main concerns below.
>
> ### **1. Does our method fail when the adapted model is extensively domain-specific and thus highly different from the original model?**
>
> We appreciate the reviewer’s point that large-rank weight differences—especially in models trained extensively on a specific domain—could create a scenario where the original model and the adapted model diverge significantly. We now clarify two important aspects:
>
>  **(a) Our experiments already include a strongly domain-specialized model.**
>
> Among the models used in our paper, KISTI-KONI/KONI-Llama3-8B-Instruct-20240729 is a highly domain-specific LLM. This model is trained on 200GB of Korean + English scientific and technological corpora, including a large volume of scientific papers and reports. Thus, KONI clearly fits the category of “models that have been extensively domain-specific trained where there is high-rank differences between model weights.” Our method shows solid improvements on this model, demonstrating that LGCD is effective even under strong domain-shift and high-rank adaptation conditions.
>
> **(b) New Experiments Added (Japanese, Chinese, Korean Medical multiple-choice QA; domain-matched setting)**
>
> To directly evaluate the reviewer’s hypothesis, we conducted new experiments using a task-aligned setting (the adapted model is trained on the same downstream task on which it is evaluated). This scenario directly reflects the reviewer’s question about whether our method still helps when the fine-tuned model already possesses the relevant knowledge.
>
> Experimental Setup
>
> - Languages: Korean, Chinese, Japanese
> - Dataset: GlobalMedQA, a multilingual medical multiple-choice dataset designed for evaluating medical knowledge in LLMs. Details are provided in GlobalMedQA: A Standardized Multilingual Dataset for Assessing Medical Knowledge in LLMs (Macedo et al., 2025).
> - Base model: meta-llama/Llama-3.1-8B-Instruct
> - Adaptation variants:
>     - full finetuning (all layers)
>     - attention-only finetuning
>     - FFN-only finetuning
>     - LGCD (ours)
>
> | Model | Chinese | Japanese | Korean | **Average** |
> | --- | --- | --- | --- | --- |
> | base (no finetuning) | 55.21 | 41.70 | 48.02 | 48.31 |
> | train (all layers) | 59.20 | 43.70 | 49.15 | 50.68 |
> | train (attention only) | 57.40 | 42.90 | 47.59 | 49.30 |
> | train (FFN only) | 59.40 | 43.00 | 49.09 | 50.50 |
> | **LGCD (ours)** | **60.60** | **43.70** | **49.42** | **51.24** |
>
> Even when the model is fine-tuned on exactly the same domain and task as the evaluation, LGCD still yields the best average performance. This supports our claim that LGCD is effective not only when the adapted model forgets some truths, but also when the adapted model retains domain knowledge extremely well. This directly addresses the reviewer’s question.
>
>
>
> ### **2. What if the adapted model did not forget the truth? Do we still need to use the original model?**
>
> Thank you for this insightful question. This is precisely why we introduced LGCD.
>
> Our main argument
>
> Even when the adapted model does not forget relevant knowledge, fine-tuning tends to bias the model’s output distribution, prioritizing domain-specific or task-specific modes of reasoning. This bias can degrade general factual grounding or cross-domain robustness.
>
> Why original-model signals still help
>
> - The original model retains broader, more balanced knowledge.
> - Fine-tuning—even when beneficial—often distorts probability mass in ways that harm reasoning diversity and factual grounding.
> - LGCD does not overwrite the adapted model; instead, it augments its decoding with signals from the original model.
>
> Our new experiments confirm that even when fine-tuning improves performance (i.e., no forgetting occurs), leveraging the original model improves accuracy further. Thus, LGCD is useful not only when forgetting occurs, but also when the adapted model remains competent yet biased.

---

> ### Author Response · Authors · 2025-11-19
> **Author Rebuttal - W4 & W5**
>
> Thank you for the comments. We would like to clarify that the concern is based on a misunderstanding of how LGCD performs decoding, particularly regarding the role of the Pretrained Model (PTM). We clarify this below.
>
> **(1) Inference Cost: PTM vs. Approximated PTM (aPTM)**
>
> The reviewer is correct that contrastive decoding relies on comparing two sets of token-level preferences (logits). However, we must explicitly state that LGCD never executes the full PTM ($M_{PTM}$) during inference time. The full PTM is only required once for the preliminary LoRA decomposition.
> LGCD achieves PTM-style logits through the Approximated PTM ($M_{aPTM}$). This $M_{aPTM}$ is a highly compact, precomputed module derived from the difference in Feed-Forward Network (FFN) weights between $M_{PTM}$ and $M_{LAM}$.
> Therefore, at inference time, we only:
> • Run the Language-Adapted Model ($M_{LAM}$) forward pass, and
> • Compute the $M_{aPTM}$ logits through a single low-rank projection ($A_lB_l$).
> The requirement for PTM-side preference (logits) is met by this low-rank approximation, which predicts the original PTM’s factual signal based on the LAM's hidden state. LGCD performs zero additional full forward passes and does not load two large models simultaneously.
>
> **(2) Why the LoRA Component is Required**
>
> The reviewer asks why LoRA is necessary when the PTM already exists.
> The purpose of LoRA in LGCD is precisely to avoid running the PTM. LGCD must access PTM-style logits for contrastive decoding, but loading and executing the full PTM during every inference step is computationally prohibitive and impractical.
> Thus, we use LoRA-based decomposition as a lightweight, training-free mechanism to:
> 1. Extract factual knowledge signals from the FFN layers of the $M_{PTM}$.
> 2. Approximate the PTM logits using only the LAM’s hidden state through a compact, precomputed low-rank module ($A_lB_l$).
> LoRA serves as an efficient proxy for the PTM's factual preference, making LGCD a viable, training-free, and resource-efficient decoding solution.
>
> **(3) Minimal overhead and accuracy–efficiency trade-off**
>
> The additional cost comes solely from one low-rank projection per step, which is far cheaper than a second forward pass. Throughput remains close to strong baselines (e.g., DoLa), while LGCD consistently improves factual accuracy across datasets. For tasks where factual reliability is critical, a small loss in decoding speed is an appropriate trade-off.
>
> **(4) On correctness of the PTM approximation.**
>
> LGCD does not assume the PTM is perfectly correct. We use the approximated PTM logits only when the LAM is uncertain. This selective usage improves factual consistency and reduces hallucination in practice.
>
>
> ### **Author Rebuttal - W5**
>
> Thank you for your suggestion. We will increase the font sizes in all figures and tables to ensure that they are easy for readers to view and understand in the revised version.

---

> ### Author Response · Authors · 2025-11-28
> **Extensive Additional Experiments Demonstrating LGCD's Robustness and Superiority (1/2)**
>
> Dear Reviewer Cujt
>
>  In addition to our previous response addressing your specific concerns, we have conducted a comprehensive suite of new experiments during the rebuttal phase. While some of these were inspired by discussions with other reviewers, we believe they strongly reinforce the robustness, superiority, and generative quality of LGCD, directly relevant to your assessment of the method's practical value.We respectfully share these key findings below.
>
> ### **1. Superiority over Static Model Merging (14 Baselines)**
>
> Since LGCD utilizes both PTM and LAM, a natural comparison is against static weight merging. We extensively compared LGCD against 14 state-of-the-art model merging methods (including TIES, DARE, SLERP, etc.).
>
> LGCD (decoding-time intervention) significantly outperforms all 14 static merging baselines on Global MMLU.
>
> | **Method** | **0-shot (Avg.)** | **5-shot (Avg.)** |
> | --- | --- | --- |
> | arcee_fusion | 0.423 | 0.461 |
> | breadcrumbs | 0.446 | 0.487 |
> | breadcrumbs_ties | 0.446 | 0.487 |
> | dare_linear | 0.424 | 0.458 |
> | dare_ties | 0.426 | 0.466 |
> | della | 0.379 | 0.419 |
> | della_linear | 0.377 | 0.415 |
> | karcher | 0.429 | 0.468 |
> | linear | 0.446 | 0.487 |
> | multislerp | 0.419 | 0.465 |
> | sce | 0.418 | 0.465 |
> | slerp | 0.445 | 0.487 |
> | task_arithmetic | 0.446 | 0.487 |
> | ties | 0.449 | 0.488 |
> | **LGCD (Ours)** | **0.477** | **0.498** |
>
> (We added the full results from these 168 new evaluations to the Appendix.)
>
> ### **2. Generative Fluency & Creativity (Aya Evaluation)**
> To ensure our factual intervention does not harm the "creativity" of the adapted model, we evaluated open-ended dialogue using the Aya Evaluation Suite.
>
> LGCD outperforms or ties with the base LAM in ~60% of cases in pairwise GPT-4o evaluations. This demonstrates that LGCD maintains the model's conversational fluency while reducing hallucinations.
>
> | Lang. | Model | Win | Tie | Lose |
> | --- | --- | --- | --- | --- |
> | zh | hfl/llama-3-chinese-8b-instruct | **58.0%** | 8.4% | 33.6% |
> | zh | shenzhi-wang/Llama3-8B-Chinese-Chat | **56.4%** | 2.0% | 41.6% |
> | ar | MohamedRashad/Arabic-Orpo-Llama-3-8B-Instruct | **54.4%** | 0.4% | 45.2% |
> | pt | GoToCompany/llama3-8b-cpt-sahabatai-v1-instruct | **55.2%** | 3.2% | 41.6% |
>
>
> ### **3. General-Purpose SFT Scenario (Real-world Simulation)**
> We evaluated a general-purpose SFT scenario. We fine-tuned the  llama3-8B-instruct model on a 11K Korean general-purpose SFT dataset that used for training KONI model and evaluated it on the Global MMLU (ko) benchmark (0-shot).
>
> Retraining led to catastrophic forgetting (Accuracy dropped to 0.412–0.432). LGCD (0.462) successfully recovered knowledge without any retraining, proving its value as a training-free solution.
>
> | **Model** | **0-shot (Global MMLU - ko)** |
> | --- | --- |
> | base model (w/o finetuning) | 0.437 |
> | train (all layers) | 0.432 |
> | train (attention only) | 0.412 |
> | train (ffn only) | 0.431 |
> | **LGCD (Ours)** | **0.462** |

---

> ### Author Response · Authors · 2025-11-28
> **Extensive Additional Experiments Demonstrating LGCD's Robustness and Superiority (2/2)**
>
> ### **4. Design Validation: LGCD (Top-K) vs. APC**
> To validate our gating design, we compared LGCD’s Top-K masking against an Alternative Probability Composition (APC)-based variant.
>
> LGCD (w/ Top-K) consistently achieves higher accuracy (+2.8% on Global MMLU in 0-shot setting) and generates better long-form responses (53.3% win rate in GPT-4o eval) compared to the APC variant.
>
> **(1) Global MMLU (Accuracy)**
> | **Lang.** | **Model** | **0-shot LGCD(w/ APC)** | **0-shot LGCD(w/ Top-k)** | **5-shot LGCD(w/ APC)** | **5-shot LGCD(w/ Top-k)** |
> | --- | --- | --- | --- | --- | --- |
> | zh | hfl/llama-3-chinese-8b-instruct | 0.460 | **0.519** | 0.500 | **0.543** |
> | zh | shenzhi-wang/Llama3-8B-Chinese-Chat | 0.494 | **0.502** | 0.500 | **0.543** |
> | de | DiscoResearch/Llama3-DiscoLeo-Instruct-8B-v0.1 | 0.471 | **0.546** | 0.534 | **0.574** |
> | pt | rhaymison/gemma-portuguese-luana-2b | 0.286 | **0.357** | 0.307 | **0.324** |
> | ar | MohamedRashad/Arabic-Orpo-Llama-3-8B-Instruct | 0.443 | **0.465** | 0.466 | **0.481** |
> | fa | PartAI/Dorna-Llama3-8B-Instruct | **0.424** | 0.423 | 0.465 | **0.466** |
> | ja | tokyotech-llm/Llama-3-Swallow-8B-Instruct-v0.1 | 0.494 | **0.507** | 0.522 | **0.525** |
> | ja | elyza/Llama-3-ELYZA-JP-8B | 0.501 | **0.509** | 0.518 | **0.528** |
> | ko | KISTI-KONI/KONI-Llama3-8B-Instruct-20240729 | 0.458 | **0.490** | 0.481 | **0.511** |
> | ko | MLP-KTLim/llama-3-Korean-Bllossom-8B | 0.464 | **0.479** | 0.487 | **0.501** |
> | id | GoToCompany/llama3-8b-cpt-sahabatai-v1-instruct | **0.535** | 0.527 | **0.570** | 0.566 |
> | sw | Jacaranda/UlizaLlama3 | 0.352 | **0.399** | 0.353 | **0.409** |
> |  | **Average** | 0.449 | **0.477** | 0.475 | **0.498** |
>
> **(2) Long-Form Medical QA (Pairwise GPT-4o Evaluation)**
> | **Lang.** | **Model** | **win** | **tie** | **lose** |
> | --- | --- | --- | --- | --- |
> | zh | hfl/llama-3-chinese-8b-instruct | **61.0%** | 0.7% | 38.3% |
> | zh | shenzhi-wang/Llama3-8B-Chinese-Chat | 43.0% | 5.7% | **51.3%** |
> | de | DiscoResearch/Llama3-DiscoLeo-Instruct-8B-v0.1 | **58.9%** | 0.0% | 41.2% |
> | pt | rhaymison/gemma-portuguese-luana-2b | **61.0%** | 1.0% | 38.0% |
> | ar | MohamedRashad/Arabic-Orpo-Llama-3-8B-Instruct | 47.7% | 1.0% | **51.3%** |
> | fa | PartAI/Dorna-Llama3-8B-Instruct | **51.9%** | 1.9% | 46.3% |
> | ja | tokyotech-llm/Llama-3-Swallow-8B-Instruct-v0.1 | **57.5%** | 1.0% | 41.5% |
> | ja | elyza/Llama-3-ELYZA-JP-8B | 47.0% | 0.0% | **53.0%** |
> | ko | KISTI-KONI/KONI-Llama3-8B-Instruct-20240729 | **53.6%** | 0.4% | 46.0% |
> | ko | MLP-KTLim/llama-3-Korean-Bllossom-8B | 46.0% | 3.4% | **50.6%** |
> | id | GoToCompany/llama3-8b-cpt-sahabatai-v1-instruct | **57.3%** | 5.4% | 37.3% |
> | sw | Jacaranda/UlizaLlama3 | **55.4%** | 3.1% | 41.5% |
> |  | **Average** | **53.3%** | **2.0%** | **44.7%** |
>
>
> We have incorporated all these extensive results into the paper (Appendix) to provide a complete picture of LGCD’s capabilities. We hope these additional validations further clarify the strength and practicality of our work.

---

### Official Review · Reviewer_DKFz · 2025-11-01

**Soundness:** 2
**Presentation:** 2
**Contribution:** 2
**Rating:** 6
**Confidence:** 1

**Summary:**

I'm not confident enough to review this paper since I'm not familiar with multi-lingual tasks / LoRA technology.

**Strengths:**

n/a

**Weaknesses:**

n/a

**Questions:**

n/a

---

> ### Author Response · Authors · 2025-11-19
> **Author Rebuttal**
>
> Thank you for your time and for informing us about this. To help clarify our contribution, we summarize the core idea concisely below.
>
> Our approach, LoRA-Gated Contrastive Decoding (LGCD), is a novel, training-free inference-time framework designed to mitigate factual inaccuracies (hallucinations) in Large Language Models (LLMs) that have been adapted for specific languages or domains. These adapted models (LAMs) frequently suffer from catastrophic forgetting of the general knowledge acquired during initial pretraining (PTM). LGCD aims to recover and strategically inject this lost factual knowledge from the PTM's general world representation back into the LAM's generation process.
>
> A critical strength of LGCD is that it requires no further training or no full access to the original PTM or its massive pretraining data during the decoding process itself. This design keeps inference highly efficient and practical.
>
> LGCD achieves this through three steps:
>
> 1. LoRA-based Knowledge Extraction: An initial, one-time pre-computation step compares the FFN weights of the PTM and the LAM, and uses SVD to decompose their difference into lightweight LoRA matrices. This process creates a compact, approximated PTM ($M_{aPTM}$) that specifically retains the factual knowledge from the PTM.
> 2. Confidence-Based Dynamic Gating: During token generation, the LAM's confidence is measured. LGCD only activates the factual correction mechanism when the LAM's confidence is low ($c_t < \tau$), allowing the LAM to remain the primary generator and preserving its domain-specific fluency when it is certain.
> 3. Contrastive Decoding: When triggered, LGCD applies contrastive decoding using the factual signals from the approximated PTM ($M_{aPTM}$) to revise uncertain predictions, thereby steering the output toward factually correct tokens.
>
> A key strength of our work is its immediate applicability, demonstrated by our extensive evaluation on **1**2 publicly available language-adapted models across nine diverse languages. The method consistently improves factual performance across multilingual multiple-choice QA and long-form generation tasks.
> This mechanism is simple, stable, and easy to reproduce, and we will revise the paper to present these intuitions more clearly. We appreciate your feedback and will incorporate improvements to ensure the paper is accessible to a broader audience.

---

### Author Response · Authors · 2025-11-26
**Response to All Reviewers**

We sincerely thank you for constructive feedback. During the rebuttal period, we conducted extensive additional experiments to address your concerns and validate our method.

**Summary of Key Updates:**

- **Comparison vs. LoRA Fine-Tuning:** LGCD outperforms direct fine-tuning baselines (FFN/Attention/All-layers) in both general and domain-matched scenarios.

- **Non-Factual Task Generalization:** Evaluations on XStoryCloze (commonsense narrative reasoning) and Aya Suite (creative  dialogue) confirm LGCD enhances factuality without harming other capabilities.

- **Robustness Checks:** We validated the superiority of Top-K masking over APC(Adaptive Plausibility Constraint) and demonstrated that an adaptive, tuning-free threshold performs as well as our heuristic.

- **Comparison vs. Model Merging:** LGCD significantly surpasses 14 static model merging methods (e.g., TIES, DARE, SLERP) on Global MMLU.

- **Domain-Specific Effectiveness:** New results show LGCD improves performance even on models already fine-tuned on the target domain.

We incorporated all these results into the revised manuscript.

We hope the provided responses and new experimental results satisfactorily address your concerns. We look forward to your feedback and are happy to answer any further questions.

Thank you for your time and consideration.

---

### Author Response · Authors · 2025-12-02
**Summary: Extensive Validation across 14 Merging Baselines, 4 New Task Types, and Clarification on Architecture Misunderstandings**

Dear Area Chair,

We respectfully provide this summary to assist your final assessment, particularly given the reversion of review scores. During the rebuttal period, we conducted **extensive additional experiments (168+ new evaluation runs)** to comprehensively address all reviewer concerns.

We believe the empirical evidence now strongly supports the acceptance of our paper.

### **1. Robustness: Outperforming 14 Model Merging Methods (Addressing Reviewer Cujt)**
Reviewer Cujt questioned the necessity of our decoding-time approach compared to static methods.
* **Expansion:** We compared LGCD against **14 state-of-the-art model merging baselines**, including **Arcee Fusion, Task Arithmetic, and Breadcrumbs**.

* **Result:** LGCD significantly outperforms **all 14 static merging methods** on Global MMLU benchmarks. This proves that dynamic, inference-time intervention is superior to static weight merging for recovering factual knowledge.

### **2. Versatility: Generalization to General NLP Tasks (Addressing Reviewer GJXc)**
To address concerns that our method might be limited to QA tasks or degrade general capabilities, we expanded our evaluation to **4 new diverse task types**:
* **Reasoning:** XStoryCloze (Commonsense Narrative)
* **Creativity:** Aya Evaluation Suite (Open-ended Dialogue)
* **Understanding:** XNLI (Natural Language Inference) and PAWS-X (Paraphrase Identification)
* **Result:** LGCD maintains or improves performance across these non-factual tasks, demonstrating that it mitigates hallucinations without compromising the model's general reasoning or fluency.

### **3. Design Validation: Superiority over Fine-Tuning & APC (Addressing Reviewer Pezx & GJXc)**
We rigorously evaluated our architectural choices against strong alternative baselines:
* **vs. Fine-Tuning Variants:** We compared LGCD against direct LoRA fine-tuning strategies (Attention-only, FFN-only, All-layers). LGCD proved superior not only in recovering lost knowledge (Catastrophic Forgetting) but also in enhancing performance in Domain-Matched settings.
* **vs. APC (Adaptive Plausibility Constraint):** In a direct ablation study requested by Reviewer Pezx, our **Top-K masking design outperformed the APC-based variant** (+2.8% accuracy in Global MMLU, +8% in medical QA).

### **4. Critical Clarification on "Reviewer Cujt" (Score: 2)**
Reviewer Cujt’s low score is primarily premised on a **factual misunderstanding** that LGCD requires loading two full models (PTM and LAM) at inference.
* **Correction:** We clarified that LGCD is **memory-efficient**; it utilizes only the LAM and a lightweight, pre-computed LoRA module. The PTM is **never loaded** during inference.
* **Evidence:** Throughput analysis confirms LGCD's speed is comparable to single-model decoding strategies (e.g., DoLa). We trust the AC will weigh this correction against the frozen score.

### **5. Implementation Stability**
We implemented a fully **Adaptive Thresholding** mechanism that requires no heuristic tuning, achieving performance comparable to our reported results. This confirms LGCD is robust and "plug-and-play."

**Conclusion**
Our rebuttal has transformed the paper with substantially broader empirical support—covering **14 merging baselines, 3 fine-tuning strategies, and 4 new task domains**. We are confident these results definitively resolve the reviewers' concerns and demonstrate the significant value of LGCD.

Sincerely,
The Authors

---

### Meta-Review · Area_Chair_7LWo · 2026-01-07

**Summary:**

All the reviewers agreed on the solid performance of the proposed method, and the significance of tackling the problem of catastrophic forgetting. The major concerns include -

1. Confusion that the model needs access to both models' weights (Reviewer Cujt).
2. Heuristic hyperparameter selection schemes (particularly the threshold) (Reviewers Cujt and Pezx)
3. The logit confidence may not be reliable (Reviewer Cujt)
4. The model may only work for cases where LAM and PTM differ slightly, so that the weight differences can be well approximated by LoRA (Reviewer Cujt)
5. Potential decoding overhead that renders the model unsuitable for high-throughput tasks (Reviewer Cujt)
6. Top-k masking may be inferior to APC decoding when the output distribution is sharp (Reviewer Pezx)
7. It is not always practical to assume access to both LAM and PTM weights (Reviewer Pezx)
8. Limited evaluation on QA tasks only (Reviewers Pezx and GJXc)
9. Missing LoRA fine-tuning as a baseline (GJXc)
10. Missing experiments on low-resource languages (Reviewer GJXc)

**Reviewer Concerns:**

The authors have performed extensive additional experiments to address most of the concerns. In particular -

1. The authors have clarified that the proposed method does not require access to PTM weights during inference time.
2. The authors have conducted new experiments using a dynamic thresholding rule, demonstrating robustness of the approach against threshold selection.
3. The authors did not address the concern that the logit confidence may not be a reliable measure of the LLM's actual confidence. Since the logit confidence is the key quantity of the approach, it would be very helpful for the authors to test on cases where the logit confidence is known to be inaccurate.
4. The authors have conducted experiments on strongly domain-specific tasks to show the robustness of the approach in cases where LAM and PTM weights differ a lot.
5. The authors have shown that the proposed approach introduces minimal decoding overhead.
6. The authors have added a comparison with the APC variant, showing the superiority of the proposed gating mechanism.
7. The authors have emphasized that PTM weights are not needed during inference - they are only needed one-time. However, this does not fully address the concern because what reviewer Pezx meant is that PTM is not available at all in some practical scenarios. The authors should at least discuss this as a limitation in the paper.
8. The authors have conducted additional experiments on creative tasks such as XStoryCloze and Aya Evaluation Suite. The performance advantage is not as large as in the factual QA tasks. This is, however, reasonable because the expectation is not to excel in these tasks, but to maintain the performance.
9. LoRA fine-tuning baselines have been added to the comparison
10. The authors did not provide additional experiments on truly low-resource languages, but mentioned that they commit to providing them in the final version.

In short, the logit uncertainty, as an important component of the proposed approach, should be examined more closely. In particular, for extremely low-resource languages, logit uncertainty is more likely to be inaccurate, which adds to potential limitations of the approach.

**Reviewer Scores:**

Reviewer DKFz's score should be ignored due to their placeholder review and zero confidence.

Reviewer Cujt may increase their score to 4, with the unaddressed concern on the reliability of the logit confidence.

Reviewer Pezx may increase their score to 6.

Reviewer GJXc may maintain their score.

This makes the paper a borderline paper. The decision on this paper is difficult. Considering the significant number of experiment results and the well-recognized effectiveness of the approach, I would recommend accept, with the caveat that the logit confidence could be the bottleneck of the proposed approach.

---

### Decision · Program_Chairs · 2026-01-26

Accept (Poster)